# Metformin alleviates stress-induced cellular senescence of aging human adipose stromal cells and the ensuing adipocyte dysfunction

Laura Le Pelletier[1†], Matthieu Mantecon[1†], Jennifer Gorwood[1], Martine Auclair[1], Roberta Foresti[2], Roberto Motterlini[2], Mireille Laforge[3], Michael Atlan[4], Bruno Fève[5], Jacqueline Capeau[1], Claire Lagathu[1*‡], Veronique Bereziat[1*‡]

[1]Sorbonne Université, Inserm UMR_S 938, Centre de Recherche Saint-Antoine (CRSA), RHU CARMMA, Institute of Cardiometabolism and Nutrition (ICAN), Paris, France; [2]University Paris-Est Créteil, INSERM, IMRB, Créteil, France; [3]CNRS, INSERM UMRS_1124, Faculté des sciences fondamentales et biomédicales, Université de Paris, Paris, France; [4]AP-HP, Tenon Hospital, Department of Plastic Surgery, Paris, France; [5]AP-HP, Saint-Antoine Hospital, Department of Endocrinology, PRISIS, Paris, France

**\*For correspondence:**
claire.lagathu@inserm.fr (CL);
veronique.bereziat@inserm.fr (VB)

†These authors contributed equally to this work
‡These authors also contributed equally to this work

**Competing interest:** The authors declare that no competing interests exist.

**Abstract** Aging is associated with central fat redistribution and insulin resistance. To identify age-related adipose features, we evaluated the senescence and adipogenic potential of adipose-derived stromal cells (ASCs) from abdominal subcutaneous fat obtained from healthy normal-weight young (<25 years) or older women (>60 years). Increased cell passages of young-donor ASCs (in vitro aging) resulted in senescence but not oxidative stress. ASC-derived adipocytes presented impaired adipogenesis but no early mitochondrial dysfunction. Conversely, aged-donor ASCs at early passages displayed oxidative stress and mild senescence. ASC-derived adipocytes exhibited oxidative stress, and early mitochondrial dysfunction but adipogenesis was preserved. In vitro aging of aged-donor ASCs resulted in further increased senescence, mitochondrial dysfunction, oxidative stress, and severe adipocyte dysfunction. When in vitro aged young-donor ASCs were treated with metformin, no alteration was alleviated. Conversely, metformin treatment of aged-donor ASCs decreased oxidative stress and mitochondrial dysfunction resulting in decreased senescence. Metformin's prevention of oxidative stress and of the resulting senescence improved the cells' adipogenic capacity and insulin sensitivity. This effect was mediated by the activation of AMP-activated protein kinase as revealed by its specific inhibition and activation. Overall, aging ASC-derived adipocytes presented impaired adipogenesis and insulin sensitivity. Targeting stress-induced senescence of ASCs with metformin may improve age-related adipose tissue dysfunction.

## Introduction

Adipose tissue is the largest lipid storage and endocrine organ in the body. Recent studies have highlighted adipose tissue's critical role in age-related diseases and metabolic dysfunction. Aging is physiologically associated with fat redistribution and a metabolic and functional decline in adipose tissue, associated with oxidative stress, inflammation, and fibrosis (*Cartwright et al., 2007*; *Kuk et al., 2009*; *Cartwright et al., 2010*; *Tchkonia et al., 2010*; *Luo and Liu, 2016*; *Stout et al., 2017*). The age-related redistribution of adipose tissue is characterized by the accumulation of truncal fat, hypertrophy of visceral adipose tissue (VAT), and overall loss of subcutaneous adipose tissue (SCAT)

(*Oikawa et al., 2016*; *Park et al., 2016*; *Mancuso and Bouchard, 2019*). An excess amount of VAT might reflect the paucity of SCAT during aging. Indeed, while accumulation of SCAT in the lower part of the body is considered to be a metabolic sink capable of buffering surplus energy, a decline in SCAT storage capacity might lead to (i) ectopic lipid deposition in the bone marrow, heart, liver, and muscles and (ii) an increase in cardiometabolic comorbidities (*Stout et al., 2017*). Accordingly, it has been suggested that appropriate SCAT plasticity and expandability guard against metabolic disorders (including insulin resistance) and that the age-related loss of these properties favors metabolic disorders (*Pasarica et al., 2009*; *McLaughlin et al., 2011*).

On the cellular level, several age-related changes in SCAT and VAT might contribute to adipose tissue dysfunction. Failure of SCAT is likely to result from the impaired recruitment of precursors and blunted adipogenesis. The identification of adipose-derived progenitor cells in the stromal vascular fraction of adipose tissue has highlighted the importance of de novo adipogenesis in adipose tissue expansion (*Cawthorn et al., 2012*). Adipose-derived stromal cells (ASCs) are defined as plastic-adherent cells expressing specific surface antigens and that are able to differentiate into osteoblasts, adipocytes, and chondroblasts in vivo and in vitro (*Dominici et al., 2006*). They can be isolated, expanded, and induced to differentiate into the above-mentioned lineages by using specific culture conditions and thus constitute a useful tool for studying age-related diseases. The abundance of adipocyte progenitors/precursors that differentiate into adipocytes is an important determinant of SCAT expandability and functionality. Adipocyte-differentiated ASCs have a crucial role in lipid handling, adipose tissue expansion, and insulin sensitivity (*Palmer and Kirkland, 2016*). With age, the frequency of ASCs decreases (*Liu et al., 2017*). A reduction in the ASCs proliferation rate might ultimately reflect cell senescence and might be involved in the onset of metabolic alterations (e.g. insulin resistance) observed during aging.

The age-dependent senescence of ASCs has been linked to a decrease in mitochondrial activity and an increase in levels of reactive oxygen species (ROS) (*Choudhery et al., 2014*; *Maredziak et al., 2016*; *Liu et al., 2017*). Although most of the literature data show that aging has a negative effect on osteoblasts and chondrocytes, the results differ with regard to the impact of senescence on the ASCs' adipogenic potential. Indeed, some studies found that aging had a negative effect on adipocyte differentiation (*Karagiannides et al., 2001*; *Murphy et al., 2002*; *Sepe et al., 2011*; *Caso et al., 2013*; *Beane et al., 2014*), whereas others found that adipocyte differentiation increased with age (*de Girolamo et al., 2009*; *Choudhery et al., 2014*; *Maredziak et al., 2016*). Lastly, it has been suggested that targeting senescent cells in adipose tissue will enhance adipogenesis and metabolic function in old age (*Xu et al., 2015*).

We show here that different mechanisms were involved in physiological, in vivo, aging and in passage-related, in vitro, aging, resulting in different adipocyte dysfunction. Thus, at early passages, ASCs isolated from SCAT of aged donors displayed stress-induced senescence leading to early adipocyte mitochondrial dysfunction, oxidative stress, and cellular insulin resistance but preserved adipogenesis. Conversely, in vitro passage-induced (long-term in vitro culture) aging in young-donor ASCs was responsible for the onset of senescence features in the absence of oxidative stress leading to a gradual decline in the proliferation and adipocyte differentiation capacity associated with enhanced insulin resistance. Interestingly, we showed that all these dysfunctions were more pronounced in aged-donors ASCs with increasing passage number.

The biguanide drug metformin is widely used to treat diabetes and also appears to modulate a number of aging-related disorders (*Barzilai et al., 2016*). Metformin exerts pleotropic effects and has a favorable influence on metabolic and cellular processes closely associated with the development of age-related conditions, such as oxidative stress, inflammation, and cellular senescence. We therefore sought to determine whether exposure of ASCs to metformin could prevent stress-induced senescence and rescue the impaired metabolic phenotype of ASCs obtained from aged donors. Our results show that, via the activation of AMP-activated protein kinase (AMPK), metformin exerted an antioxidant effect and improved mitochondria metabolism. Therefore, metformin alleviated the stress-induced cellular senescence of aged-donor ASCs and rescued adipocyte differentiation and function, which might be involved in the drug's insulin-sensitizing effect in vivo.

# Results

## In vitro aging results in senescence and mitochondrial dysfunction while physiological aging induces early oxidative stress in ASCs

In order to determine the impact of aging on ASCs, we evaluated three cellular models: (i) ASCs isolated from human SCAT samples obtained from young adults (under the age of 25- referred as 'young-donor') that were cultured in vitro from early (P3) to late passage (P11); (ii) physiologically aged ASCs that were isolated from older adults (over the age of 60- referred as 'aged-donor') at early passage, (iii) and, finally, aged-donor ASCs cultured from P3 to P11.

First, we observed that the in vitro increased passage number of young ASCs led to a senescent phenotype. The population doubling times (PDT) of young-donor ASCs at P7 and P11 were slightly longer than at P3 (*Figure 1A–B*). The mean percentage of senescent young-donor ASCs (i.e. those positive for senescence-associated (SA)-β-galactosidase) was increased at P7 and P11 (*Figure 1C–D*) although the level of lysosome accumulation measured by Lysotracker fluorescence remained unchanged (*Figure 1E*). Accordingly, young-donor ASCs expressed significantly greater levels of the cell cycle arrest proteins p16$^{INK4}$ and p21$^{WAF1}$ at P7 and P11 and of the senescence-associated protein prelamin A at P11 compared to P3 (*Figure 1F–G*). However, the level of ROS production was the same regardless of the passages (*Figure 1H*), indicating that oxidative stress was not enhanced in that setting. In order to evaluate the presence of mitochondrial dysfunction, we measured both the volume (by using Mitotracker that labels mitochondria) and the membrane potential (with the aggregate to monomer ratio of the JCI dye) of mitochondria. Young-donor ASCs did not show any modification of mitochondrial mass over the passages (*Figure 1I*) but presented a decreased mitochondrial membrane potential, mostly at P11 as compared to P3 (*Figure 1J*), in favor of late mitochondrial dysfunction. These data suggest that long-term in vitro culture of young-donor ASCs induced the onset of senescence with mitochondrial dysfunction but without enhanced oxidative stress.

At P3, aged-donor and young-donor ASCs had similar proliferative abilities and thus a similar PDT (*Figure 1A–B*). Aged-donor ASCs presented a mild senescent phenotype as compared to young-donor ASCs, characterized by a slight increase in senescent cell count, greater lysosome accumulation, and higher expression of p16$^{INK4}$, p21$^{WAF1}$, and prelamin A (*Figure 1C–G*). This mild senescence was not associated with mitochondrial dysfunction (*Figure 1I–J*). In agreement, a Seahorse experiment performed in young-donor and aged-donor ASCs at P3 did not reveal any difference in basal oxygen consumption rate (OCR) and maximal respiration measured after the addition of the uncoupling agent carbonyl cyanide 4-trifluoromethoxyphenylhydrazone (FCCP) (*Figure 2*). It is noteworthy that, unlike young-donor ASCs, aged-donor ASCs presented early oxidative stress (*Figure 1H*).

Finally, increased cell passages of aged-donor ASCs resulted in both physiological and in vitro aging with a more pronounced senescent phenotype regardless of the passage analyzed. In contrast to young-donor ASCs, the PDT for aged-donor ASCs increased markedly from P5 to P11 (*Figure 1A–B*), indicating a steady increase in growth inhibition. It is important to note that cell viability did not change significantly over the culture period (data not shown). Aged-donor ASCs also presented a marked increase in SA-β-galactosidase activity (*Figure 1C–D*), lysosome accumulation (*Figure 1E*), p16$^{INK4}$, p21$^{WAF1}$, and prelamin A expression (*Figure 1F–G*) at P7 and P11. Moreover, oxidative stress and mitochondrial dysfunctions were amplified over the passages (*Figure 1H–I*).

Taken as a whole, these results show that only physiological aging led to senescence associated with oxidative stress and that during long-term in vitro culture, aged-donor ASCs expressed senescence earlier and more intensely than young-donor ASCs did.

## In vitro senescence is responsible for altered adipocyte differentiation, while physiological aging is associated with preserved differentiation but dysfunction in adipocyte-differentiated ASCs

We next evaluated the impact of in vitro and physiological aging on the ASCs' ability to differentiate in vitro into mature adipocytes. At P3, P7, or P11, confluent cells were induced to differentiate for 14 days in a pro-adipogenic medium.

Increasing in vitro passage was associated with lower lipid accumulation in adipocytes differentiated from young-donor ASCs (*Figure 3A–B*) via a decreased expression of the pro-lipogenic and pro-adipogenic markers SREBP1c and PPARγ (*Figure 3C–D*). Adipocytes differentiated from young-donor

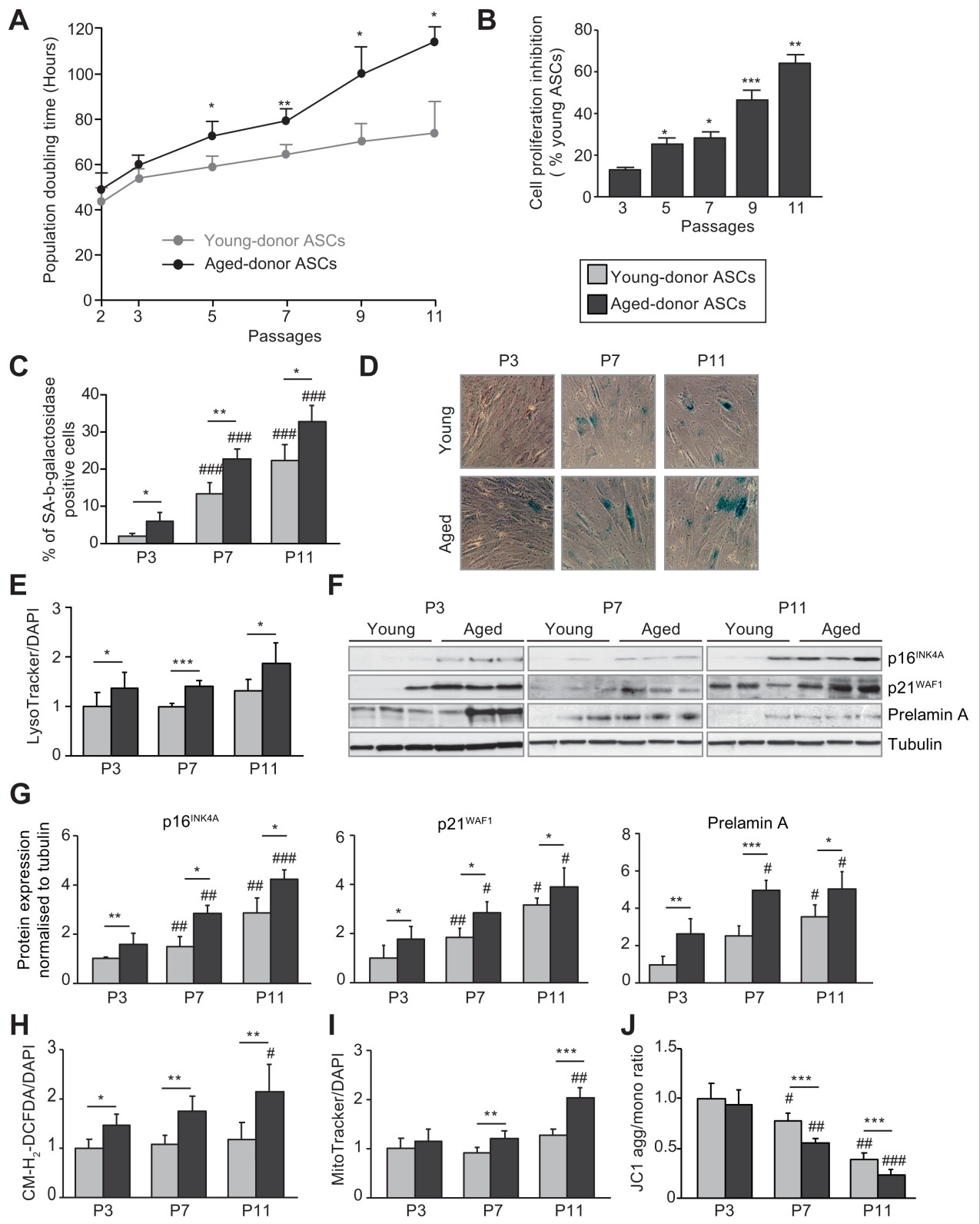

**Figure 1.** In vivo and in vitro aging are differently associated with senescence, oxidative stress, and mitochondrial dysfunction in cultured adipose-derived stromal cells (ASCs). After isolation from the abdominal subcutaneous adipose tissue (SCAT) of young (gray circles or bars) and aged (black circles or bars) donors, ASCs were cultured from passages (P) 3–11. (**A**). Calculation of the mean ± standard error of the mean (SEM) population doubling time (PDT) is described in the Materials and methods section. Times were determined at the indicated passage (n = 9, in triplicate). (**B**)

*Figure 1 continued on next page*

*Figure 1 continued*

The % inhibition of cell proliferation was calculated for aged-donor ASCs by determining the increase in total cell number that occurred after 7 days, compared to young-donor ASCs. (**C**) Senescence was evaluated in terms of senescence-associated (SA)-β-galactosidase activity and expressed as the proportion (in %) of SA-β-galactosidase-positive cells at pH 6 in aged-donor ASCs vs. young-donor ASCs at the same passage (at P3, P7, and P11). (**D**) Representative micrographs of SA-β-galactosidase-positive cells. (**E**) Lysosomal accumulation (normalized against 4',6-diamidino-2-phenylindole dihydrochloride [DAPI]) was assessed with the Lysotracker fluorescence probe and expressed as the fluorescence ratio for aged-donor ASCs vs. young-donor ASCs at P3. (**F**) Whole-cell lysates of aged-donor and young-donor ASCs at P3, P7, and P11 were analyzed by immunoblotting. Representative immunoblots of the cell cycle arrest markers p16INK4A and p21WAF1, prelamin A, and tubulin (the loading control) from three donors in each group are shown. (**G**) Quantification of western blot was normalized to young-donor ASCs at P3. (**H**) Reactive oxygen species production (normalized against DAPI) was assessed by the oxidation of CM-H2DCFDA and expressed as a ratio relative to young-donor ASCs at P3. (**I**) Mitochondrial mass (normalized against DAPI) was evaluated with Mitotracker Red-Probe and expressed as a ratio relative to young-donor ASCs at P3. (**J**) The cationic dye JC1 was used to evaluate the mitochondrial membrane potential. The results are expressed as the ratio of aggregate/monomer fluorescence. Results are quoted as the mean ± SEM. *$p < 0.05$, **$p < 0.01$, ***$p < 0.001$ for aged- vs. young-donor ASCs, #$p < 0.05$, ##$p < 0.01$, ###$p < 0.001$ vs. young- or aged-donor ASCs at P3. All experiments were performed in triplicate with ASCs isolated from four different donors in each group.

The online version of this article includes the following source data for figure 1:

**Source data 1.** SA-ß-galactosidase activity at P3.

**Source data 2.** SA-ß-galactosidase activity at P7.

**Source data 3.** SA-ß-galactosidase activity at P11.

**Source data 4.** Analysis by western blot of senescence markers p16 p21 and prelamin A.

ASCs also presented a decrease in insulin-induced phosphorylation of Akt, a key enzyme involved in short-term metabolic responses to insulin, in favor of a cellular insulin resistance at P7 and P11 (*Figure 3I*).

Interestingly, in vitro aging of young-donor ASCs, that resulted in ASC senescence, was not associated with early mitochondrial dysfunction in derived adipocytes (at P7, *Figure 3F and G*), thus disentangling ASC senescence and adipocyte mitochondrial dysfunction.

Interestingly, at P3, adipocytes differentiated from young-donor or aged-donor ASCs had similar adipogenic capacities as shown by the similar triglyceride accumulation (*Figure 3A–B*) and pro-lipogenic and pro-adipogenic markers expression level (*Figure 3C–D*). By contrast, only adipocytes differentiated from aged-donor ASCs displayed enhanced oxidative stress (*Figure 3E*) and mitochondrial dysfunction characterized by increased mitochondrial mass and decreased membrane potential (*Figure 3F–G*). Moreover, the analysis of basal OCR, using Seahorse experiments, did not reveal any significant difference (*Figure 4*). However, the adipocytes differentiated from aged-donors ASCs displayed a major decrease in their maximal respiratory capacity, as indicated by a significantly lower increase in OCR after the addition of the uncoupling agent FCCP (*Figure 4*). Therefore, increased

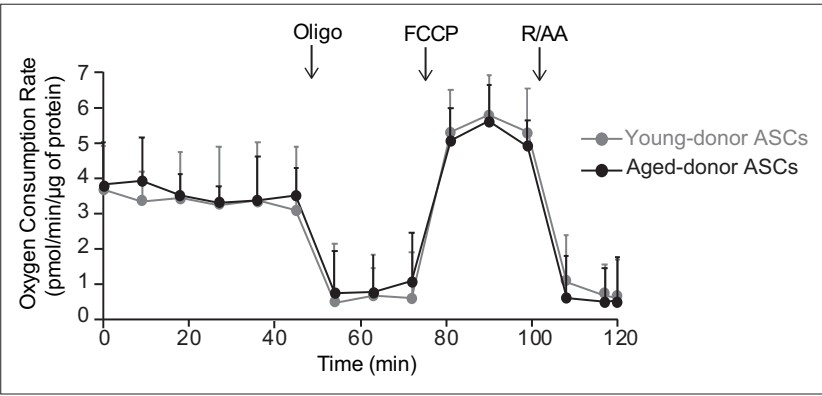

**Figure 2.** The oxygen consumption rate of young-donor or aged-donor adipose-derived stromal cells (ASCs) is not different at early passage. Cellular respiration was assessed by the measurement of the oxygen consumption rate (OCR) in young-donor (gray circles) and aged-donor ASCs (black circles) at P3 using Seahorse. OCR was measured at baseline and after sequential addition of oligomycin (oligo: inhibitor of ATP synthase), carbonyl cyanide 4-tr ifluoromethoxyphenylhydrazone (FCCP: uncoupling agent), and rotenone/antimycin A (R/AA, inhibitors of the respiratory chain complexes I and III). Results are quoted as the mean ± standard error of the mean (SEM). The experiment was performed in triplicate with ASCs isolated from three different donors in each group.

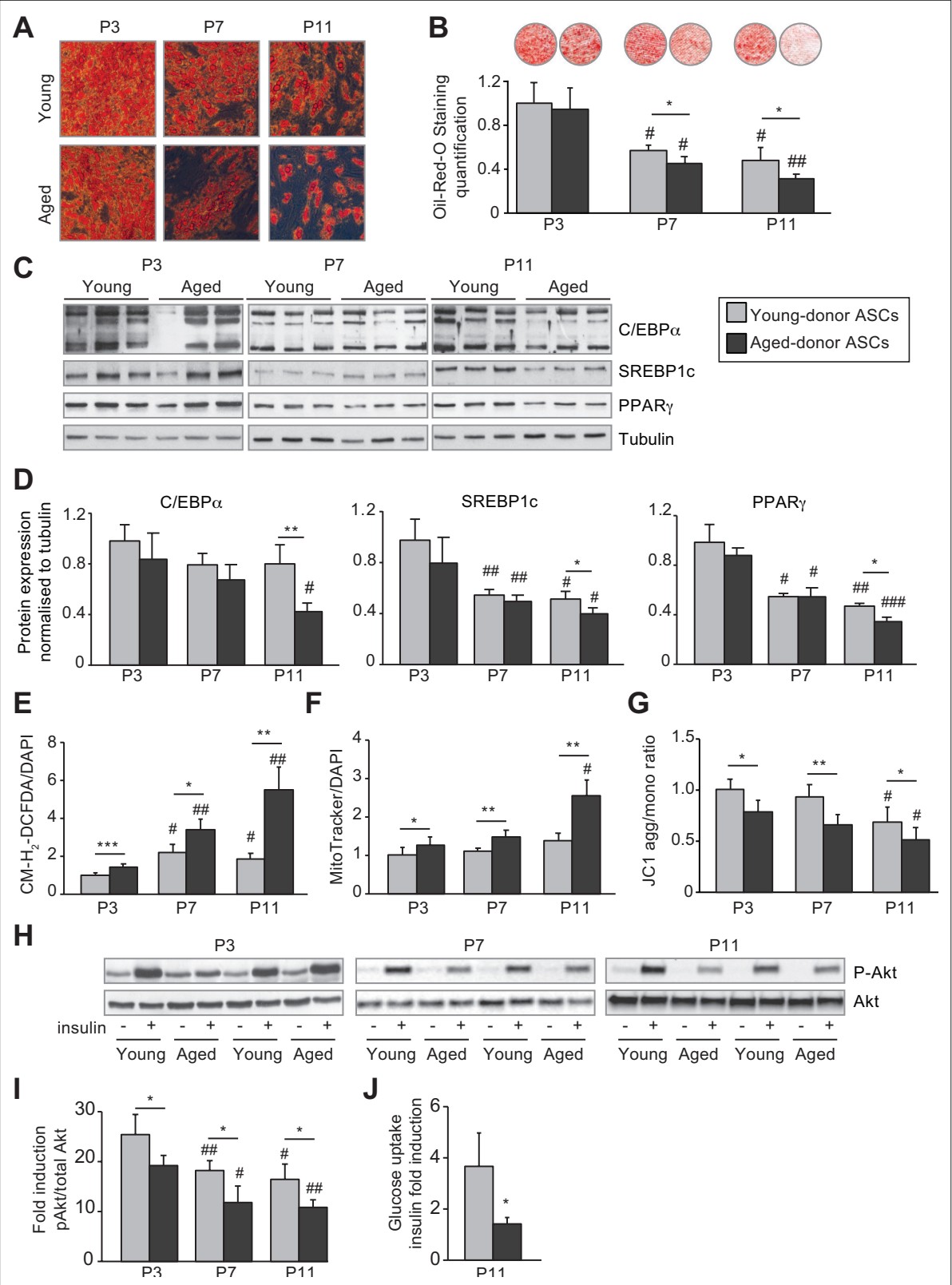

**Figure 3.** In vivo and in vitro aging are differently associated with adipose dysfunction of adipocytes differentiated from young-donor and aged-donor adipose-derived stromal cells (ASCs). The ASCs were differentiated into adipocytes for 14 days at P3, P7, and P11. (**A**) Cells were stained with Oil-Red-O to visualize lipid droplets 14 days post-induction, and representative micrographs are shown. (**B**) Quantification of Oil-Red-O staining of adipocytes differentiated from ASCs and representative scans of wells are shown. (**C**) Whole-cell lysates, at day 14 post-induction, of adipocytes differentiated from

*Figure 3 continued on next page*

*Figure 3 continued*

ASCs isolated from young and aged donors, cultured until P3, P7, and P11, were analyzed by immunoblotting. Representative immunoblots of C/EBPα, SREBP-1c, PPARγ, and tubulin (loading control) are shown. (**D**) Quantification of western blot was normalized to young-donor ASCs at P3. (**E**) Reactive oxygen species (ROS) production, normalized against 4′,6-diamidino-2-phenylindole dihydrochloride (DAPI). (**F**) Mitochondrial mass (normalized against DAPI) and (**G**) mitochondrial membrane potential were assessed in adipocytes derived from aged-donor ASCs as described in *Figure 1*. (**H**) Whole-cell lysates (extracted at day 14 post-induction) of adipocytes differentiated from young- and aged-donor ASCs and stimulated (or not) with insulin were analyzed with immunoblotting. Representative immunoblots of Akt and phospho-Akt (Ser473) are shown. (**I**) The phosphorylated Akt/total Akt ratio was determined in a densitometric analysis. (**J**) Insulin sensitivity at P11 in adipocytes differentiated from young- and aged-donor ASCs was evaluated by measuring glucose uptake in basal and insulin-stimulated conditions as described in the Materials and methods section. The insulin fold induction was determined. Results are quoted as the mean ± standard error of the mean (SEM). *$p < 0.05$, **$p < 0.01$, ***$p < 0.001$ for aged-donor vs. young-donor ASCs, #$p < 0.05$, ##$p < 0.01$, ###$p < 0.001$ vs. young- or aged-donor ASCs at P3. All experiments were performed in duplicate or triplicate with ASCs isolated from four different donors in each group.

The online version of this article includes the following source data for figure 3:

**Source data 1.** Analysis by western blot of sadipogenic markers C/EBPa, SREBP1C and PPARg.

oxidative stress associated with mild senescence in aged-donor ASCs resulted in mitochondrial dysfunction in ASC-derived adipocytes, suggesting that adipocyte mitochondrial dysfunction could be linked to increased oxidative stress in ASCs.

Adipocytes differentiated from aged-donor ASCs also presented decreased acute insulin-induced phosphorylation of Akt indicating of insulin resistance. Throughout the passages, adipocytes differentiated from aged-donor ASCs presented altered adipogenic capacities, oxidative stress, and mitochondrial dysfunction that were more pronounced as compared to adipocytes differentiated from young-donor ASCs (*Figure 3A–G*). Insulin resistance paralleled oxidative stress (*Figure 3H–I*). Altogether, these results support the hypothesis that the senescence of ASCs, observed during the long-term in vitro culture, could be responsible for the altered adipogenic differentiation. By contrast, the physiological aging that leads to oxidative stress in aged-donors ASCs at early passages would lead to enhanced metabolic alterations in adipocytes, but preserved adipocyte differentiation capacities.

To further delineate the link between oxidative stress and senescence of ASCs and of derived adipocytes, we induced senescence of young-donor ASCs at P3 by treating them with HIV protease inhibitors (PIs) and analyzed the impact of this induced senescence on adipocyte-differentiated ASCs. PIs are known to induce senescence by accumulating farnesylated prelamin A, responsible for premature aging syndromes (*Hernandez-Vallejo et al., 2013*, *Afonso et al., 2016*, *Afonso et al., 2017*). As shown in *Figure 5*, a treatment for 30 days with lopinavir associated with a low dose of ritonavir (LPV/r) led to ASC senescence (*Figure 5A*) but also to marked oxidative stress (*Figure 5B*) and mitochondrial

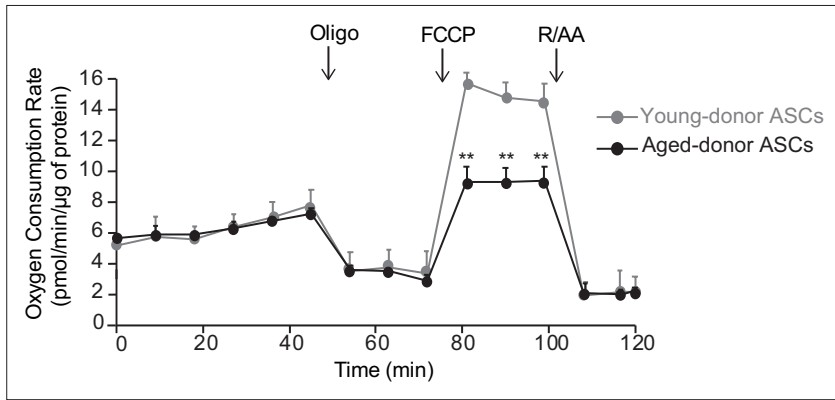

**Figure 4.** The oxygen consumption rate (OCR) of adipocyte differentiated from young-donor or aged-donor adipose-derived stromal cells (ASCs) is different at early passage. Cellular respiration was assessed by the measurement of OCR in adipocytes differentiated from young-donor (gray circles) and aged-donor ASCs (black circles) at P3, using Seahorse, at baseline and after sequential addition of oligomycin (oligo), 4-trifluoromethoxy phenylhydrazone (FCCP), and rotenone/antimycin A (R/AA). Results are quoted as the mean ± standard error of the mean (SEM). **$p < 0.01$, adipocytes differentiated from young-donor ASCs vs. adipocytes differentiated from aged-donor ASCs. The experiment was performed in triplicate with ASCs isolated from three different donors in each group.

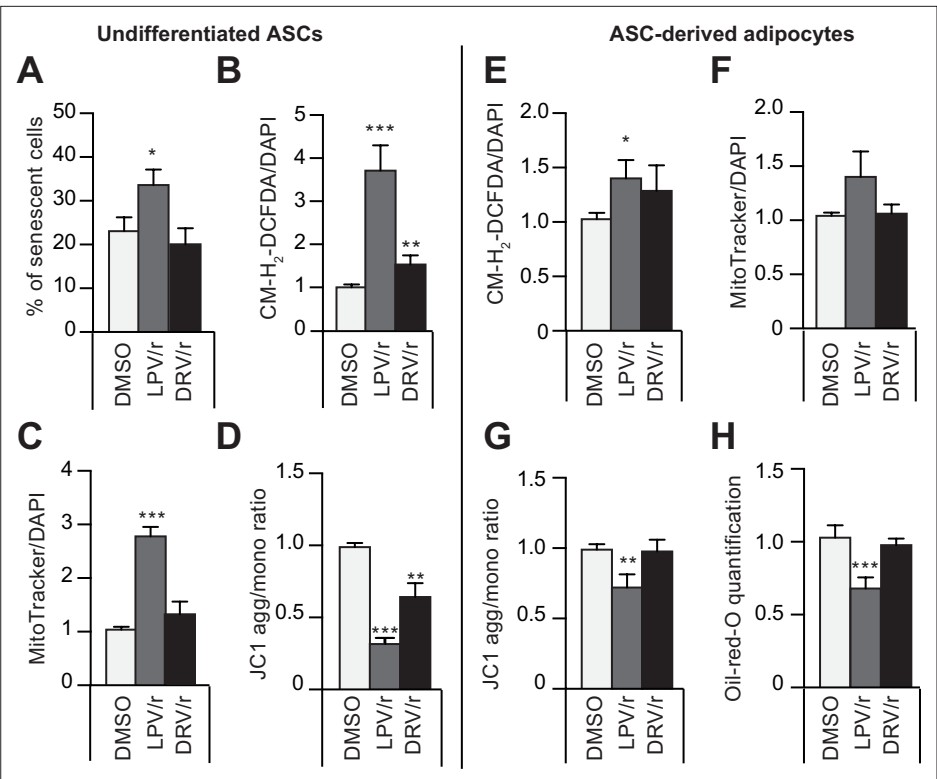

**Figure 5.** The protease inhibitors (PIs) lopinavir/r and darunavir/r induce, at different extents, in young-donor adipose-derived stromal cells (ASCs) at early passage senescence, oxidative stress, and mitochondrial dysfunction leading or not to altered ASC-derived adipocytes. ASCs were treated during 30 days with DMSO or PIs: lopinavir (LPV) or darunavir (DRV) associated with a low dose of ritonavir (LPV/r or DRV/r) (**A–D**). The ASCs were then differentiated into adipocytes for 14 days in the absence of PIs (**E–H**). (**A**) ASCs' senescence was evaluated in terms of senescence-associated (SA)-β-galactosidase activity and expressed as the proportion (in %) of SA-β-galactosidase-positive cells at pH 6. (**B**) Reactive oxygen species (ROS) production (normalized against 4',6-diamidino-2-phenylindole dihydrochloride [DAPI]) was assessed by the oxidation of CM-H2DCFDA and expressed as a ratio relative to DMSO. (**C**) Mitochondrial mass (normalized against DAPI) was evaluated with Mitotracker Red-Probe and expressed as a ratio relative to DMSO. (**D**) The cationic dye JC1 was used to evaluate the mitochondrial membrane potential. In ASC-derived adipocytes, 14 days post-induction, (**E**) ROS production (**F**) mitochondrial mass, and (**G**) mitochondrial membrane potential normalized to DAPI were evaluated. (**H**) Cells were stained with Oil-Red-O to visualize lipid droplets. Results are quoted as the mean ± standard error of the mean (SEM). *p < 0.05, **p < 0.01, ***p < 0.001 vs. DMSO. All experiments were performed in triplicate with ASCs isolated from three different donors in each group.

The online version of this article includes the following source data for figure 5:

**Source data 1.** Impact of HIV protease inhibitors on SA-ß-galactosidase activity.

dysfunctions (*Figure 5C–D*). Adipocytes derived from these LPV/r-treated ASCs exhibited increased oxidative stress (*Figure 5E*) and mitochondrial dysfunctions (*Figure 5F–G*) together with altered adipogenesis (*Figure 5H*). This confirms that induction of oxidative stress and senescence in young-donor ASCs results in severe adipose dysfunction in derived adipocytes. Interestingly, another PI with less adverse effects, darunavir associated with a low dose of ritonavir (DRV/r), did not increase senescence in ASCs but induced mild oxidative stress and mitochondrial dysfunction. Accordingly, treatment of ASCs with DRV/r did not impede differentiation of derived adipocytes further indicating that mild oxidative stress in the absence of senescence in ASCs did not impair adipogenesis.

## Metformin prevents the onset of senescence and associated dysfunctions in ASCs isolated from aged donors but not in ASCs from young donors

Next, we evaluated whether metformin could alleviate cellular senescence in ASCs. To that end, ASCs were treated with metformin from P3 to P11. As shown in *Figure 6A–B*, metformin did not modify the PDT of young-donor ASCs and had no effect on the level of senescence or oxidative stress and a mild effect on mitochondrial dysfunction (*Figure 6C–I*).

Conversely, metformin prevented the above-mentioned relative decrease in cell proliferation in aged-donor ASCs presenting stress-induced senescence (*Figure 6A–B*). Accordingly, metformin rescued the higher percentage of senescent cells (*Figure 6C–D*), the greater lysosome accumulation (*Figure 6E*), and the higher expression of cell cycle inhibitors p21$^{WAF1}$ and p16$^{INK4}$ (*Figure 6F*) previously observed at P11.

Taken as a whole, these data show that in aged-donor ASCs, metformin reversed the dysfunction associated with oxidative stress-induced senescence. Indeed, metformin treatment reversed oxidative stress and mitochondrial dysfunction to the levels observed in young-donor ASCs whereas it did not reverse senescence features in young-donor ASCs which do not display enhanced oxidative stress (*Figure 6G–I*).

## Metformin restores the ability of aged-donor but not young-donor ASCs to differentiate into adipocytes

We checked whether metformin pre-treatment was able to rescue the altered adipogenesis potential of young-donor and aged-donor ASCs. At first, ASCs were treated with metformin, then adipogenesis was induced at P11 in the absence of the drug to bypass its anti-adipogenic impact (*Marycz et al., 2016*; *Chen et al., 2018*) (*Figure 7—figure supplement 1*). First, we observed that metformin was unable to rescue the altered adipogenesis observed in adipocytes derived from young-donor ASCs at P11 (*Figure 7A–B*). Conversely, metformin restored the adipogenic capacity of aged-donor ASCs at P11 to the level observed in young-donor ASCs. Indeed, metformin treatment during the proliferation state increased lipid accumulation (*Figure 7A–B*) and increased C/EBPα, and SREBP1c expression (*Figure 7C–D*). Lastly, metformin pre-treatment lowered levels of oxidative stress (*Figure 7G*), leading to a partial rescue of the insulin sensibility in adipocytes-differentiated ASCs (*Figure 7E–F*) but did not restore mitochondrial function (*Figure 7H–I*).

## The beneficial effect of metformin on ASC senescence is mediated by AMPK activation

We hypothesized that, in mechanistic terms, metformin's action might be based on (among other things) the phosphorylation and thus activation of AMPK. As shown in *Figure 8A*, metformin treatment was associated with (i) greater AMPK expression in both young-donor and aged-donor ASCs but (ii) greater AMPK phosphorylation in aged-donor ASCs only. These findings are in line with metformin's beneficial effect on aged-donor ASCs and lack of effect on young-donor ASCs, as reported above.

To confirm AMPK's potential role in the action of metformin, we determined whether the drug's effects were influenced by treatment with the AMPK inhibitor compound C. Indeed, we observed that in aged-donor ASCs, treatment with compound C counteracted the beneficial effect of metformin on cell proliferation (*Figure 8B*), senescence marker levels (*Figure 8C–D*), oxidative stress (*Figure 8E*), and mitochondrial dysfunction (*Figure 8F–G*). Interestingly, adipocytes differentiated from aged-donor ASCs, then pre-treated with both metformin and compound C during proliferation but not during differentiation did not show metformin's beneficial effect on adipogenesis (*Figure 8H*) and insulin-stimulated glucose uptake (*Figure 8I*). Accordingly, we showed that AMPK constitutive activation with AICAR (5-aminoimidazole-4-carboxyamide ribonucleoside), an agonist of the AMPK pathway (*Merrill et al., 1997*), induced a decrease of aged-donor ASCs senescence and mitochondrial dysfunction at P11 (*Figure 8—figure supplement 1*) leading to increased triglycerides and decreased ROS production in adipocytes differentiated from aged-donor ASCs (*Figure 8—figure supplement 2*). These findings highlighted AMPK's role in the beneficial action of metformin on aged-donor ASCs and the derived adipocytes.

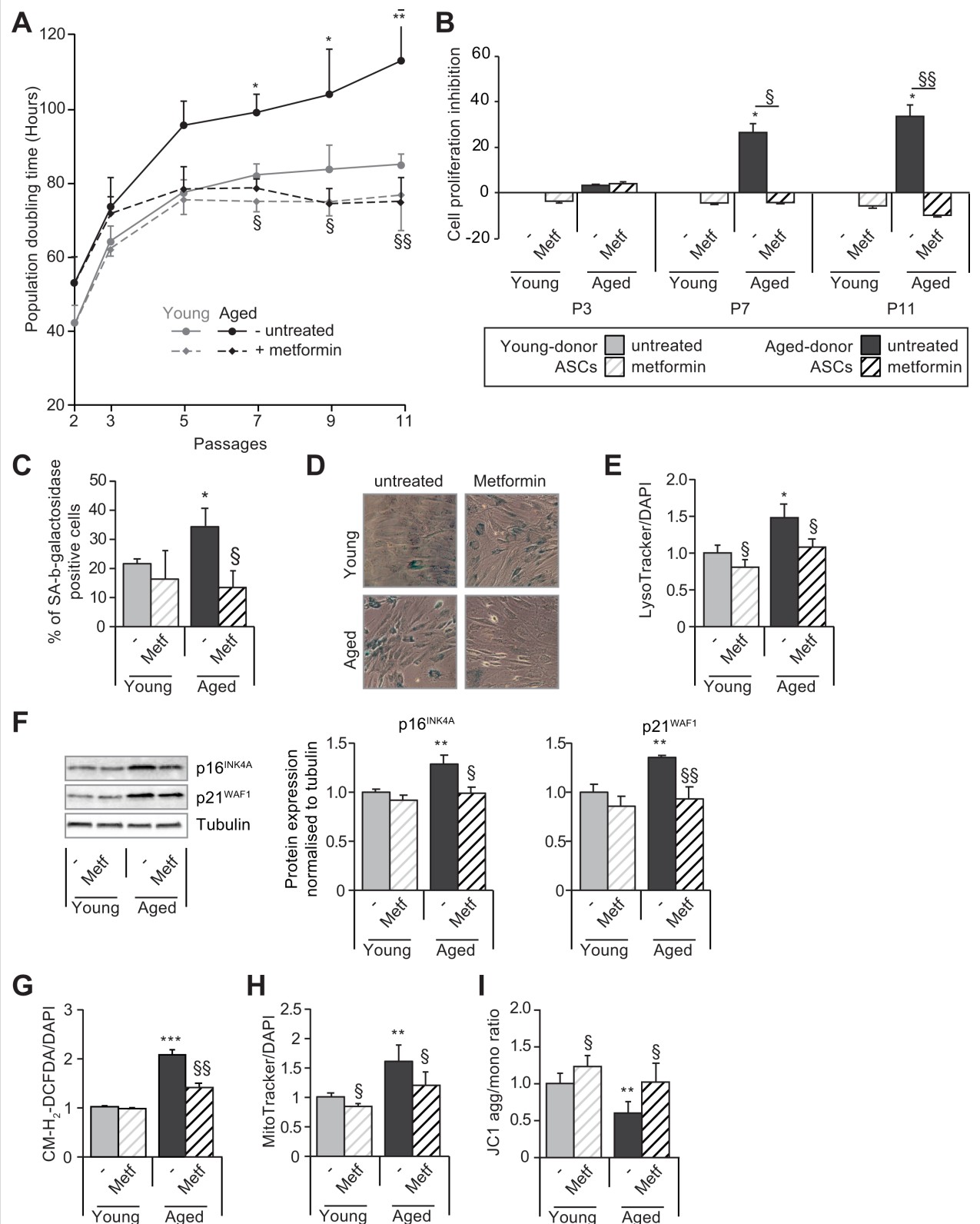

**Figure 6.** Metformin partially prevents the senescence and associated dysfunctions in adipose-derived stromal cells (ASCs) obtained from aged donors but not in those obtained from young donors. Metformin (25 µmol/L) was added to the culture medium of aged-donor and young-donor ASCs from P3 onward (young-donor ASC: gray dots and bars in the absence of metformin, gray dotted lines or gray striped bars in the presence of metformin; aged-donor: black circles and bars in the absence of metformin, black dotted lines and striped bars in the presence of metformin). Mean population doubling

*Figure 6 continued on next page*

*Figure 6 continued*

times (PDT) were determined at the indicated passages in aged-donor and young-donor ASCs treated (or not) with metformin at the same passage. (**B**) The % inhibition of cell proliferation was calculated for aged-donor ASCs and young-donor ASCs treated by metformin by determining the increase in total cell number that occurred after 7 days, compared to young-donor ASCs. (**C**) Senescence was evaluated in terms of senescence-associated (SA)-β-galactosidase activity and expressed as the proportion (in %) of SA-β-galactosidase-positive cells at pH 6 in metformin-treated ASCs vs. non-treated ASCs at P11. (**D**) Representative micrographs of SA-β-galactosidase positive cells. (**E**) Lysosomal accumulation (normalized against 4',6-diamidino-2-phenylindole dihydrochloride [DAPI]) was assessed with the Lysotracker fluorescence probe in metformin-treated ASCs vs. non-treated ASCs at P11. (**F**) Whole-cell lysates of aged-donor and young-donor ASCs treated (or not) with metformin were analyzed at P11 by immunoblotting. Representative immunoblots of the cell cycle arrest markers p16INK4A and p21WAF1 and of tubulin (the loading control) for two donors in each group are shown. Quantitation of western blots, normalized against the values for non-treated young-donor ASCs at P11. (**G**) Reactive oxygen species (ROS) production, (**H**) mitochondrial mass (both normalized against DAPI) and (**I**) mitochondrial membrane potential were assessed as described in *Figure 1* in metformin-treated ASCs vs. non-treated ASCs at P11. The results correspond to the mean ± standard error of the mean (SEM). *$p < 0.05$, **$p < 0.01$, ***$p < 0.001$ for aged-donor vs. young-donor ASCs, §$p < 0.05$, §§$p < 0.01$ metformin-treated vs. non-treated ASCs. All experiments were performed in duplicate or triplicate in ASCs isolated from three different donors in each group.

The online version of this article includes the following source data for figure 6:

**Source data 1.** Impact of metformin on SA-ß-galactosidase activity.

**Source data 2.** SA-ß-galactosidase activity in aged and young untreated ASCs.

**Source data 3.** Impact of metformin on p16 p21 protein expression.

## Discussion

Our study shows for the first time that passage-related (in vitro) or physiological (age of the donor, in vivo) aging leads to different senescence mechanisms and thus differently affects adipocyte and adipose tissue function. In vitro passage-induced senescence impedes adipocyte differentiation while stress-induced senescence (as observed in physiological aging) results in oxidative stress and mitochondrial dysfunction but not impaired differentiation in ASC-derived adipocytes. When aged-donor ASCs age in vitro, the two processes are present resulting in a severe dysfunction in derived adipocytes. By activating AMPK, metformin restored almost all the features of aging in aged-donor ASCs to the levels observed in young-donor ASCs. However, metformin did not modify these variables in the latter cells.

Our data are in line with previous studies of human ASCs (*Zhu et al., 2009*; *Jung et al., 2019*; *Alicka et al., 2020*). The ASCs have a typical mesenchymal stem cell morphology, and there are little or no morphological differences between young-donor and aged-donor cells in early passages. In most studies, the PDT of human ASCs did not vary with age (*Zhu et al., 2009*; *Chen et al., 2012*; *Ding et al., 2013*). Accordingly, we did not observe any differences between young-donor and aged-donor ASCs with regard to the phenotype or proliferation rate early in the period of culture (P3). However, with increasing passage numbers, aged-donor cells developed severe features of senescence (such as increased SA-β-galactosidase activity, oxidative stress, and mitochondrial dysfunctions) earlier than young-donor cells that also present in vitro features of senescence. These alterations might have driven the progressive fall in proliferation capacity seen for aged-donor ASC from P5 onward. Indeed, we found that aged-donor ASCs had molecular, morphological, and functional impairments late in culture (P11). We propose that our results provide a better understanding of the molecular events occurring in ASCs in the course of aging.

With regard to the mechanisms that might promote senescence in ASCs, greater mitochondrial dysfunction and ROS production have previously been linked to the occurrence of age-dependent ASC senescence. Decreased mitochondrial function and elevated mitochondrial generation of ROS are thought to be critical in the aging process (*Seo et al., 2010*; *Correia-Melo et al., 2016*). Interestingly, ROS accumulation was observed at P3 in aged-donor ASCs, whereas mitochondrial dysfunctions were only observed at P7 suggesting that ROS could initially not originate from mitochondria. In agreement, Seahorse experiment did not show any difference in respiration capacities between aged- and young-donor ASCs. We report here that increased senescence during in vitro aging was not associated with early mitochondrial dysfunction in derived adipocytes while mild senescence and increased oxidative stress observed in the situation of in vivo aging were associated with mitochondrial dysfunction. Therefore, we propose to disentangle ASC senescence from adipocyte mitochondrial

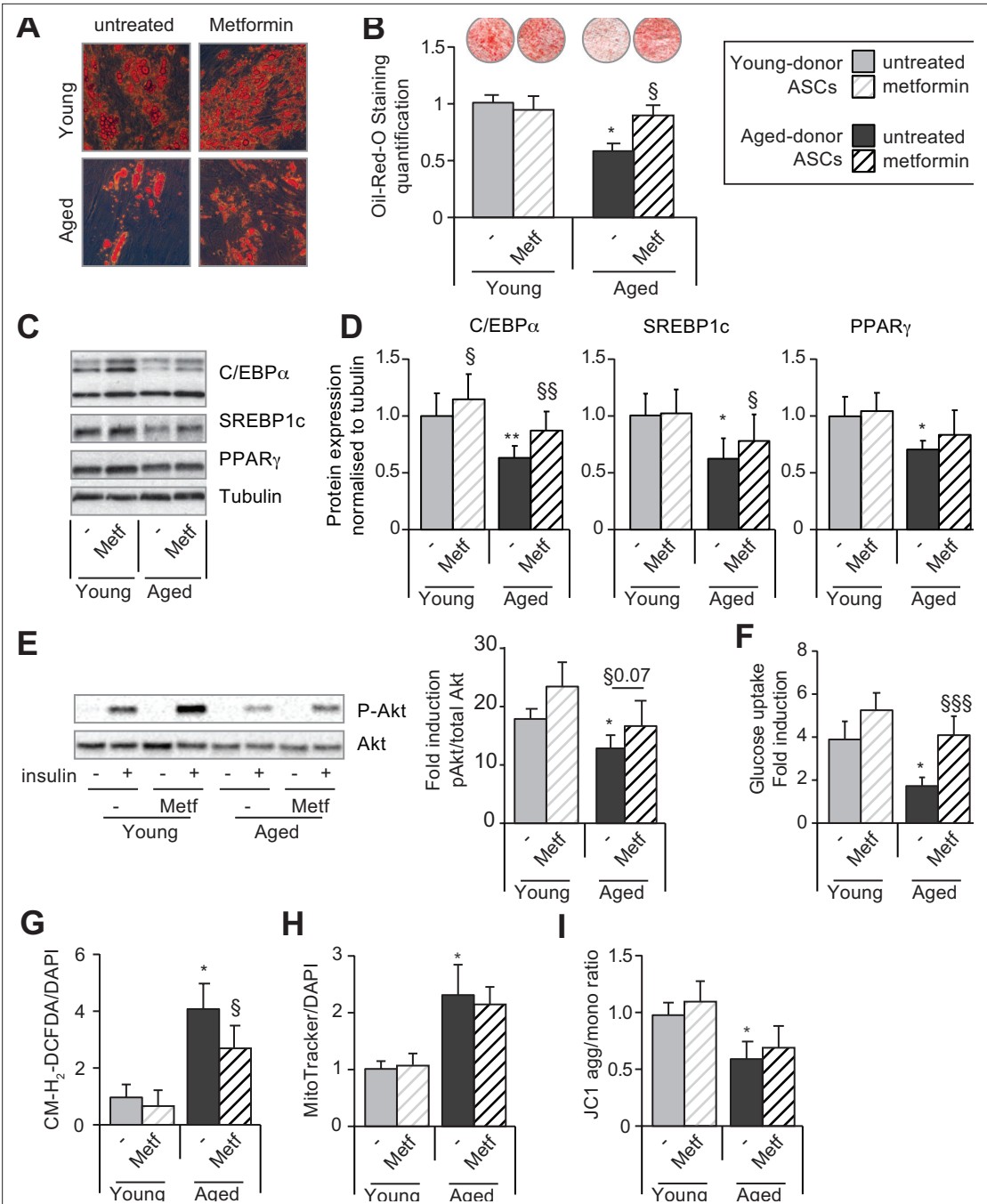

**Figure 7.** Metformin can improve the adipogenic differentiation capacity of aged-donor but not young-donor adipose-derived stromal cells (ASCs). Metformin was added to the culture medium of young- and aged-donor ASCs from P3 onward. The ASCs were differentiated into adipocytes for 14 days at P11, in the absence of metformin (young-donor ASC: gray bars in the absence of metformin, gray striped bars in the presence of metformin; aged-donor: black bars in the absence of metformin, black striped bars in the presence of metformin). (**A**) Cells were stained with Oil-Red-O to visualize lipid droplets 14 days post-induction, and representative micrographs are shown. (**B**) Quantification of Oil-Red-O staining and representative scans of wells are shown. (**C**) Whole-cell lysates on day 14 post-induction from adipocytes differentiated from non-treated young-donor and aged-donor ASCs at P11 and treated (or not) with metformin were analyzed by immunoblotting. Representative immunoblots of C/EBPα, SREBP-1c, PPARγ, and tubulin (the loading control) are shown. (**D**) Quantification of western blots is shown. (**E**) Whole-cell lysates extracted at day 14 post-induction stimulated (or not) by insulin from differentiated ASCs were analyzed by immunoblotting. Representative immunoblots of Akt and phospho-Akt (Ser473) and quantification of the pAkt/Akt are shown. (**F**) Insulin sensitivity was evaluated at P11 in adipocytes differentiated from non-treated young-donor and aged-donor ASCs treated (or not) with metformin, by measuring the glucose uptake in response to insulin and calculating the insulin fold induction, as described in the Materials and methods section. (**G**) Reactive oxygen species (ROS) production, (**H**) mitochondrial mass, and (**I**) mitochondrial membrane potential (both

*Figure 7 continued on next page*

*Figure 7 continued*

normalized against 4',6-diamidino-2-phenylindole dihydrochloride [DAPI]) were assessed as described in *Figure 1*. The results are expressed as the mean ± standard error of the mean (SEM). *p < 0.05 for adipocytes differentiated from aged-donor vs. adipocytes differentiated from young-donor ASCs, §p < 0.05, §§p < 0.01, §§§p < 0.001 metformin-treated vs. non-treated adipocytes differentiated from ASCs. All experiments were performed in duplicate on cells isolated from three different donors in each group.

The online version of this article includes the following source data and figure supplement(s) for figure 7:

**Source data 1.** Impact of metformin on C/EBPa, SREBP1C and PPARg protein expression.

**Figure supplement 1.** Culture scheme indicating that adipose-derived stromal cells (ASCs) were cultured in the presence or not of metformin along the passages but differentiated in adipocytes in the absence of metformin in the culture medium.

dysfunction and, conversely, to associate ASC increased oxidative stress with adipocyte mitochondrial dysfunction.

It has already been reported that ASC senescence impairs adipogenic differentiation capacity. In particular, several other studies of human and murine mesenchymal stromal cells had shown a reduction in differentiation potential upon aging in vitro (*Choudhery et al., 2014*, *Maredziak et al., 2016*). Our data are in agreement with this point. Moreover, they allow to better decipher the respective roles of oxidative stress and senescence regarding adipogenesis. Accordingly, we show that in vitro passage-related senescence impairs adipogenesis even in the absence of enhanced oxidative stress. Conversely, in physiological aging, adipocyte-differentiated ASCs present early enhanced oxidative stress and mitochondrial dysfunctions but not reduced adipogenic potential. Thus, senescence, but not oxidative stress alone, impairs adipogenic differentiation. However, a high level of oxidative stress results in stress-induced senescence and in turn in reduced adipogenic differentiation. When young-donor ASCs at early passages, that is, devoid of any age-related alteration, were treated with lopinavir, both oxidative stress and senescence occurred resulting in impaired adipogenesis. Conversely, darunavir induced neither senescence nor impaired adipogenesis.

One of the study's objectives was to determine whether preventing the onset of senescence in aged-donor ASCs restored their adipogenic capacity. There is experimental evidence that metformin extends lifespan in model organisms, sparking interest in its clinical relevance (*Martin-Montalvo et al., 2013*). We used a metformin concentration of 25 µmol/L, in the upper range observed in diabetic patients and considered as a safe concentration (*Frid et al., 2010*). As a senomorphic drug, it has been shown that metformin can reduce the SA secretory phenotype. Accordingly, we observed that metformin decreased activin A secretion, a molecule that has been previously shown to impair adipogenesis (*Xu et al., 2015*), in both young-donor and aged-donor ASCs regardless the passage (data not shown). However, in our model, metformin did not restore the adipogenic capacity of senescent young-donor ASCs at P11, suggesting that (i) metformin could act by another mechanism, (ii) metformin was not able to reverse the ASCs' senescence features when oxidative stress is low, and (iii) metformin was able to reverse stress-induced senescence by acting on oxidative stress.

Metformin was shown to decrease ROS production by ASCs (*Marycz et al., 2016*) and to significantly improve ASC proliferation and function, in correlation with a higher mitochondrial membrane potential (*Smieszek et al., 2019*). Here, we showed that aged-donor ASCs expressed and activated AMPK to a lower extent than young-donor ASCs, and that this impairment could be reversed by metformin treatment. This effect is related to metformin's ability to inhibit complex 1 in the mitochondrial electron transport chain, leading to activation of AMPK and a reduction in endogenous ROS production. Accordingly, we found that metformin was a potent antioxidant and prevented the onset of mitochondrial dysfunction in aged-donor ASCs. Interestingly, we found that metformin treatment had a beneficial effect on aged-donor ASCs but not on young-donor ASCs; this suggests that in vitro and in vivo aging have different mechanisms, which result in different cellular and molecular alterations. It has been shown that metformin has various effects on several pathways and targets, in addition to AMPK (*Barzilai et al., 2016*). Even though metformin's ability to inhibit mitochondrial complex 1 has been best characterized, many other pathways are affected. Nonetheless, we observed that beneficial effects of metformin were lost in the presence of compound C (a potent AMPK inhibitor) or mimicked with the AMPK activator AICAR

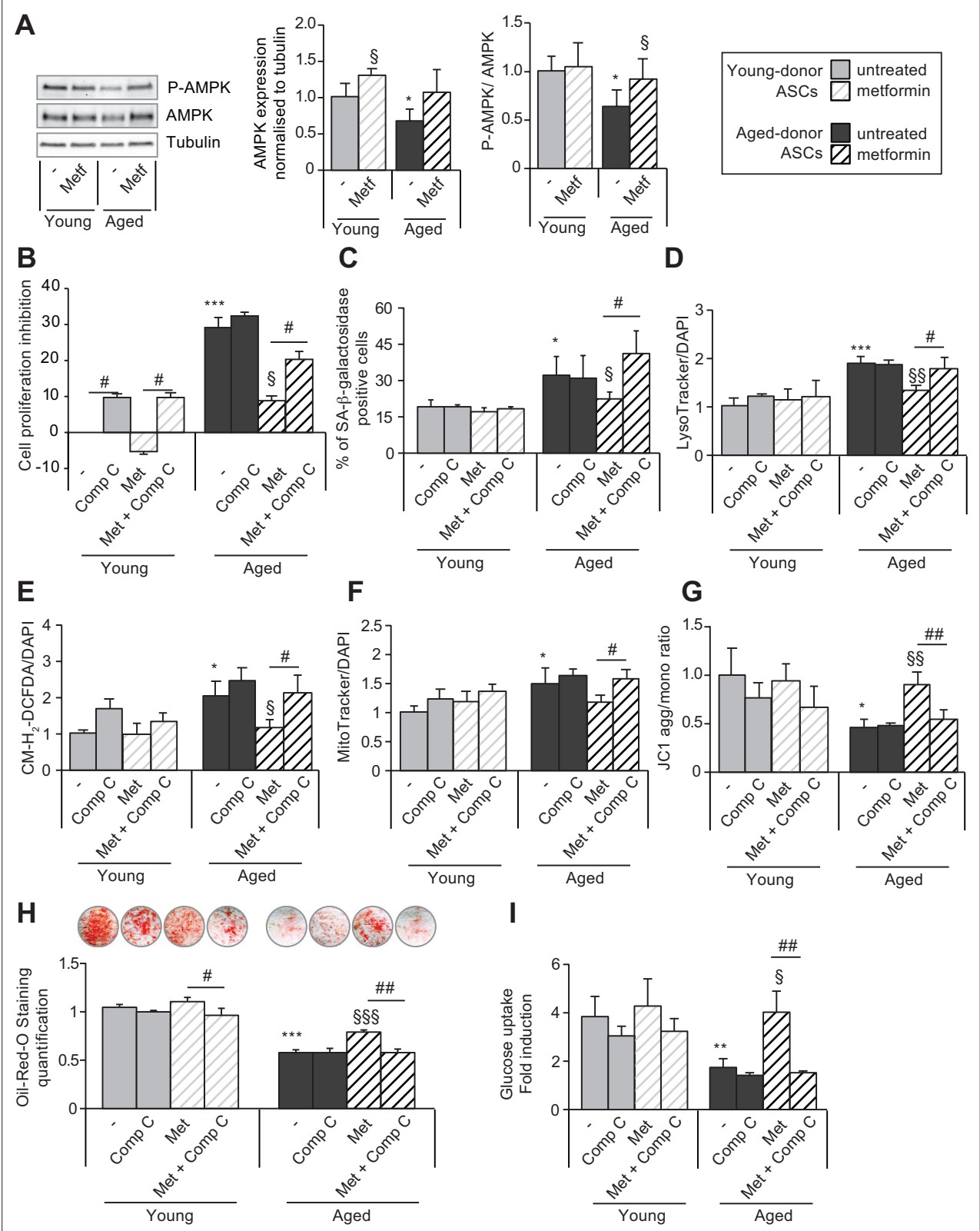

**Figure 8.** The beneficial effects of metformin on aged-donor adipose-derived stromal cell (ASC) senescence is mediated by AMP-activated protein kinase (AMPK) activation. Metformin was added to the culture medium of aged-donor and young-donor ASCs from P3 onward. To evaluate the role of AMPK activation, compound C was added at P11. After 7 days of treatment, the experiments on ASCs were carried out (young-donor ASC: gray bars in the absence of metformin or in the presence of compound C, gray striped bars in the presence of metformin or metformin and compound C; aged-

*Figure 8 continued on next page*

*Figure 8 continued*

donor: black bars in the absence of metformin or in the presence of compound C, black striped bars in the presence of metformin or metformin and compound C). (**A**) Whole-cell lysates of aged-donor and young-donor ASCs treated (or not) with metformin and compound C at P11 were analyzed by immunoblotting. Representative immunoblots of AMPK, phospho-AMPK, and tubulin (the loading control) and a graph quantifying AMPK (normalized against tubulin) and the pAMPK/AMPK ratio are shown. (**B**) The % inhibition of cell proliferation was calculated for aged-donor ASCs and young-donor ASCs treated or not with metformin or compound C, by determining the increase in total cell number that occurred after 7 days, compared to young-donor ASCs. (**C**) Senescence was evaluated in terms of senescence-associated (SA)-β-galactosidase activity and was expressed as described in *Figure 1*. (**D**) Lysosomal accumulation (normalized against 4′,6-diamidino-2-phenylindole dihydrochloride [DAPI]) was assessed with the Lysotracker fluorescence probe. (**E**) Reactive oxygen species (ROS) production, (**F**) mitochondrial mass (both normalized against DAPI), and (**G**) mitochondrial membrane potential were assessed as described in *Figure 1*. (**H**) The ASCs were then differentiated into adipocytes on P11 in the absence of metformin and compound C. Cells were stained with Oil-Red-O to visualize lipid droplets 14 days post-induction. Quantification of Oil-Red-O staining and representative scans of wells are shown. (**I**) Insulin sensitivity was evaluated at P11 in adipocytes differentiated from non-treated young-donor and aged-donor ASCs treated (or not) with metformin and/or compound C, by measuring the glucose uptake in response to insulin and calculating the insulin fold induction, as described in the Materials and methods section. Results are expressed as the mean ± standard error of the mean (SEM). *$p < 0.05$, **$p < 0.01$ for aged-donor vs. young-donor ASCs, §$p < 0.05$, §§$p < 0.01$ for metformin-treated ASCs vs. non-treated ASCs. #$p < 0.05$, ##$p < 0.01$ for compound C and metformin-treated ASCs vs. metformin-treated ASCs. All experiments were performed in duplicate or triplicate on ASCs isolated from at three different donors in each group.

The online version of this article includes the following source data and figure supplement(s) for figure 8:

**Source data 1.** Impact of Compound C on SA-ß-galactosidase activity in aged ASCs.

**Source data 2.** Impact of Compound C on SA-ß-galactosidase activity in young ASCs.

**Source data 3.** Impact of metformin on AMPK protein expression.

**Figure supplement 1.** 5-Aminoimidazole-4-carboxyamide ribonucleoside (AICAR) can reverse senescence and associated dysfunctions in aged-donor adipose-derived stromal cells (ASCs).

**Figure supplement 1—source data 1.** SA-ß-galactosidase activity in aged and young untreated ASCs.

**Figure supplement 1—source data 2.** SA-ß-galactosidase activity in aged and young metformin treated ASCs.

**Figure supplement 1—source data 3.** SA-ß-galactosidase activity in aged and young AICAR treated ASCs.

**Figure supplement 1—source data 4.** Impact of AICAR on AMPK protein expression.

**Figure supplement 2.** 5-Aminoimidazole-4-carboxyamide ribonucleoside (AICAR) can restore the adipogenic differentiation capacity of senescent aged-donor adipose-derived stromal cells (ASCs).

– highlighting the central role of the AMPK-linked signaling network in the ASC aging process (*Salminen and Kaarniranta, 2012*; *Burkewitz et al., 2014*).

To the best of our knowledge, the present study is the first to have shown that metformin can partially rescue/maintain the adipogenic capacity of aged-donor ASCs. Metformin has been shown to inhibit adipogenesis (*Marycz et al., 2016*; *Chen et al., 2018*) when cells are exposed during the process of differentiation. Here, we used a novel model in which ASCs were only treated with metformin before the induction of differentiation. By removing metformin during ASC differentiation, we were able to rule out a direct effect on adipogenesis. Thus, our data suggest that metformin's prevention of oxidative stress-induced senescence and dysfunction during ASC proliferation restored adipogenesis. Interestingly, we also observed that metformin increased the insulin sensitivity of newly differentiated ASCs isolated from aged-donor ASCs by activating AMPK; this activation has been shown to contribute to the drug's insulin-sensitizing effect in vivo – particularly in the liver and in muscle (*Salminen and Kaarniranta, 2012*).

Several studies strongly suggest that metformin could improve adipose tissue function. A treatment with metformin leads, in most cases, to fat loss and improved insulin sensitivity, but these effects are largely related to actions outside adipose tissue. A few studies evaluated the effect of metformin on the different fat depots. In the large clinical study 'Diabetes Prevention Program Research group', both SCAT and VAT were reduced in glucose-intolerant women after 1 year of treatment with metformin (*Fujimoto et al., 2007*). As well, adiponectin level was mildly improved while leptin was decreased in favor of improved adipose tissue function (*Goldberg et al., 2019*). Improved adipose tissue function is also observed in SCAT explants isolated from obese subjects treated ex vivo with metformin but not in isolated adipocytes suggesting that metformin affected cells from the stroma vascular fraction, that contains adipose stromal cells, and not directly differentiated adipocytes (*Zulian et al., 2011*). Finally, metformin improved expression of genes involved in adipogenesis and triglyceride synthesis, arguing for its ability to improve adipose function during aging (*Kulkarni et al., 2018*).

The present study had several limitations. First, we used ASCs obtained from abdominal SCAT; ASCs obtained from other subcutaneous or VAT depots might not behave in the same way. Another limitation concerns the use of fibroblast growth factor 2 (FGF2), which is required for the prolonged in vitro expansion of ASCs. It has been shown that FGF2-treated ASCs are larger, less readily proliferative, and more senescent than control ASCs (*Cheng et al., 2020*). Thus, we cannot rule out a possible masking effect of FGF2 on the aging of ASCs. Finally, we cannot address in our ASC model the potential role of the cross talk with immune cells that could also affect adipose tissue functions.

Aging is associated with the induction of senescence and associated disorders (including impaired adipogenesis, oxidative stress, and insulin resistance) in ASCs isolated from SCAT. These changes might be involved in the alterations in fat redistribution (particularly a paucity of SCAT, favoring VAT accumulation) and cardiometabolic diseases observed during aging. Our results suggest that metformin may be a promising candidate for treating age-related dysfunction in adipose tissue (*Figure 9*) and thus warrants evaluation in a clinical setting.

# Materials and methods

## Key resources table

| Reagent type (species) or resource | Designation | Source or reference | Identifiers | Additional information |
|---|---|---|---|---|
| Antibody | Anti-tubulin (Mouse monoclonal) | Merck – Sigma-Aldrich | T5168, RRID:AB_477579 | WB (1:10 000) |
| Antibody | Anti-P21 (Mouse monoclonal) | BD Bioscience | 554262, RRID:AB_395331 | WB (1:1000) |
| Antibody | Anti-P16 (Mouse monoclonal) | BD Bioscience | 551154, RRID:AB_394078 | WB (1:1000) |
| Antibody | Anti-prelamin A (Goat polyclonal) | Santa Cruz Biotechnology | sc-6214, RRID:AB_648150 | WB (1:1000) |
| Antibody | Anti-C/EBPA (Rabbit polyclonal) | Santa Cruz Biotechnology | sc-61, RRID:AB_631233 | WB (1:500) |
| Antibody | Anti-PPARG (Rabbit polyclonal) | Santa Cruz Biotechnology | sc-7196, RRID:AB_654710 | WB (1:500) |
| Antibody | Anti-SREBP1 (Rabbit polyclonal) | Santa Cruz Biotechnology | sc-366, RRID:AB_2194229 | WB (1:500) |
| Antibody | Anti-goat IgG, HRP-linked | Santa Cruz Biotechnology | sc-2354, RRID:AB_628490 | WB (1:3000) |
| Antibody | Anti-Phospho-AMPK (Thr172) (Rabbit monoclonal) | Cell Signaling | 4188, RRID:AB_2169396 | WB (1:1000) |
| Antibody | Anti-AMPK (Rabbit polyclonal) | Cell Signaling | 2532, RRID:AB_330331 | WB (1:1000) |
| Antibody | Anti-Akt (Rabbit polyclonal) | Cell Signaling | 9272, RRID:AB_329827 | WB (1:1000) |
| Antibody | Anti-Phospho-Akt (Ser473) (Rabbit monoclonal) | Cell Signaling | 9271, RRID:AB_329825 | WB (1:1000) |
| Antibody | Anti-rabbit IgG, HRP-linked | Cell Signaling | 7074, RRID:AB_2099233 | WB (1:3000) |
| Antibody | Anti-mouse IgG, HRP-linked | Cell Signaling | 7076, RRID:AB_330924 | WB (1:3000) |
| Chemical compound, drug | X-Gal 40 mg/mL DMF | Euromedex | UX-1000–05 | |
| Chemical compound, drug | Oligomycine | Merck – Sigma-Aldrich | O4876 | |
| Chemical compound, drug | Carbonyl cyanide-*p*-trifluoromethoxyphenylhydrazone (FCCP) | Merck – Sigma-Aldrich | C2920 | |
| Chemical compound, drug | Rotenone | Merck – Sigma-Aldrich | R8875 | |
| Chemical compound, drug | Antimycine A | Merck – Sigma-Aldrich | A8674 | |
| Chemical compound, drug | Metformin hydrochloride | Merck – Sigma-Aldrich | PHR1084 | |
| Chemical compound, drug | Dimethyl sulfoxide | Merck – Sigma-Aldrich | D8418 | |
| Chemical compound, drug | 3-Isobutyl-1-methylxanthine | Merck – Sigma-Aldrich | I5879 | |

*Continued on next page*

*Continued*

| Reagent type (species) or resource | Designation | Source or reference | Identifiers | Additional information |
|---|---|---|---|---|
| Chemical compound, drug | Dexametasone | Merck – Sigma-Aldrich | D4902 | |
| Chemical compound, drug | Rosiglitazone | Merck – Sigma-Aldrich | R2408 | |
| Chemical compound, drug | Collagenase B | Merck – Sigma-Aldrich | 11088815001 | |
| Chemical compound, drug | Formalin Solution, Neutral Buffered | Merck – Sigma-Aldrich | HT-501 | |
| Chemical compound, drug | Oil-Red-O | Merck – Sigma-Aldrich | O9755 | |
| Chemical compound, drug | Isopropanol | Merck – Sigma-Aldrich | 563935 | |
| Chemical compound, drug | Lopinavir | Santa Cruz Biotechnology | sc-207831 | |
| Chemical compound, drug | Ritonavir | Santa Cruz Biotechnology | sc-208310 | |
| Chemical compound, drug | Darunavir | Santa Cruz Biotechnology | sc-218079 | |
| Chemical compound, drug | AICAR | Santa Cruz Biotechnology | sc-200659 | |
| Chemical compound, drug | Compound C | Santa Cruz Biotechnology | sc-361173 | |
| Chemical compound, drug | DAPI | Thermo Fisher Scientific | D1306 | |
| Chemical compound, drug | CM-H2-DCFDA | Thermo Fisher Scientific | C6827 | |
| Chemical compound, drug | MitoTacker | Thermo Fisher Scientific | M7512 | |
| Chemical compound, drug | JC-1 | Thermo Fisher Scientific | T3168 | |
| Chemical compound, drug | Lysotracker | Thermo Fisher Scientific | L7526 | |
| Commercial assay or kit | Glucose Uptake-Glo Assay | Promega | J1341 | |
| Peptide, recombinant protein | Human Insulin | Merck – Sigma-Aldrich | I9278 | |
| Peptide, recombinant protein | Human FGF-basic | Peprotech | 100-18B | |
| Software, algorithm | Image J | Image J | RRID:SCR_003070 | |
| Software, algorithm | Prism | Prism | RRID:SCR_005375 | |

## Isolation, culture, and treatment of ASCs

The human SCAT samples from which ASCs were isolated were obtained from 10 healthy women undergoing plastic surgery. The women were young adults (n = 5; mean ± standard error of the mean [SEM] age: 21.2 ± 3.2 years; mean ± SEM BMI: 23.6 ± 1.2 kg/m$^2$) or aged adults (n = 5; mean ± SEM age: 60.0 ± 0.7 years; mean ± SEM BMI: 25.4 ± 0.4 kg/m$^2$) (see *Table 1* for details).

Before surgery, all donors provided informed written consent to use of their tissue specimens for research purposes. The study was performed in compliance with the principles of the Declaration of Helsinki and was approved by the local independent ethics committee. The ASCs were isolated using collagenase B (0.1%) (Roche Diagnostics, Basel, Switzerland) digestion technique, as described previously (*Gorwood et al., 2019*; *Gorwood et al., 2020*). After centrifugation, stromal vascular fraction was filtered, rinsed, and plated (10,000 cells/cm$^2$) in Eagle's Minimum Essential Medium

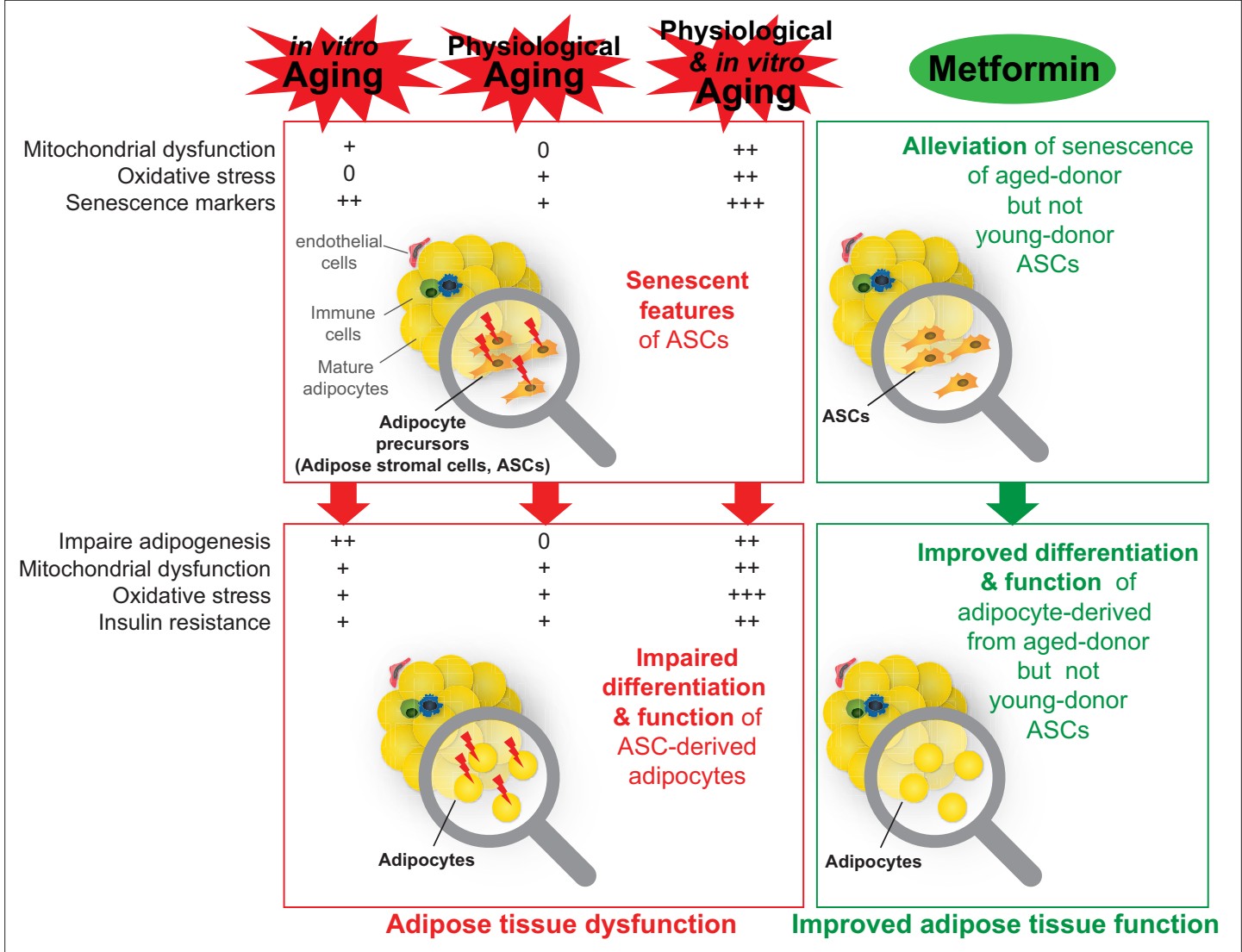

**Figure 9.** A recapitulative scheme of the alterations observed in adipose stromal cells undergoing in vitro, physiological, and both in vitro and physiological aging and in derived adipocytes. Metformin alleviated stress-induced senescence of aged-donor adipose-derived stromal cells (ASCs) resulting in improved adipogenesis and insulin sensitivity to the levels observed in young-donor ASCs.

alpha (Thermo Fisher Scientific, Courtaboeuf, France) supplemented with 10 % fetal bovine serum (FBS) (PAN-Biotech, Aidenbach, Germany), 2 mmol/L glutamine, 100 U/mL penicillin/streptomycin, 10 mmol/L HEPES (Thermo Fisher Scientific), and 2.5 ng/mL FGF2 (PeproTech, Rocky Hill, NJ). On average, we isolated 65,000 cells per mL of SCAT liposuction sample, with similar yield between SCAT from aged or young donors. Upon confluence, adherent cells were trypsinized (Thermo Fisher Scientific) and seeded at a density of 2000/cm$^2$ for the study of proliferating ASCs and of 4000 cells/cm$^2$ prior to the induction of adipogenesis. During expansion, cells were exposed (or not) to 25 µmol/L metformin (Merck, Sigma-Aldrich, St Quentin Fallavier, France). The metformin concentration chosen in our study is close to Cmax observed in elderly subjects, which exhibited higher average Cmax than younger subjects (*Jang et al., 2016*). At a late passage (P11), cells were exposed (or not) to 0.1 µmol/L of compound C (Merck, Sigma-Aldrich). Proliferating ASCs were exposed for 30 days (from P3 to P9) to DMSO (0.01%) or the PIs: lopinavir (LPV/r 10 µmol/L) or darunavir (DRV/r 10 µmol/L) in association with 2 µmol/L ritonavir (Merck, Sigma-Aldrich), at clinically relevant concentrations near Cmax (*Hernandez-Vallejo et al., 2013*). Proliferating ASCs isolated from young and aged donor were cultured at P11 with or without AICAR (100 mmol/L) (Santa Cruz Biotechnology) for 1 week and analyzed as described in *Figure 8—figure supplement 1*. ASCs were induced to differentiate

**Table 1.** Main characteristics of subcutaneous adipose tissue (SCAT) donors used for adipose-derived stromal cell (ASC) isolation.

| Sex | Age (years) | BMI (kg/m²) | Ethnicity |
|-----|-------------|-------------|-----------|
| F | 29 | 23.5 | Caucasian |
| F | 16 | 23.7 | Caucasian |
| F | 16 | 26.6 | Caucasian |
| F | 16 | 24.6 | Caucasian |
| F | 29 | 19.3 | Caucasian |
| F | 61 | 26.0 | Caucasian |
| F | 62 | 26.6 | Sub-Saharan African |
| F | 58 | 24.9 | Caucasian |
| F | 60 | 24.4 | Caucasian |
| F | 59 | 24.6 | Caucasian |

into adipocyte differentiation for 14 days without AICAR. Upon day 14, adipocyte-derived ASCs were analyzed as described in *Figure 8—figure supplement 2*.

## Adipocyte differentiation

Differentiation of ASCs was induced by the addition of pro-adipogenic Dulbecco's modified Eagle medium (DMEM), 4.5 g/L glucose (Thermo Fisher Scientific), 10 % FBS, 2 mmol/L glutamine, 100 U/mL penicillin/streptomycin, 10 mmol/L HEPES, 1 µmol/L dexamethasone, 250 µmol/L 3-isobutyl-1-methylxanthine (IBMX), 1 µmol/L rosiglitazone, 1 µmol/L insulin (Merck, Sigma-Aldrich) for 5 days, and then maintained in DMEM with rosiglitazone and insulin until day 14. Cells were then stained for Oil-Red-O (Merck, Sigma-Aldrich) and quantified at 520 nm as described previously (*Gorwood et al., 2019*; *Gorwood et al., 2020*). To analyze the impact of ASC senescence and oxidative stress on adipogenesis, adipocytes were differentiated in the absence of metformin (*Figure 7—figure supplement 1*).

## Cellular proliferation and senescence

Cellular senescence was evaluated in terms of the cell PDT at each cell passage, as described previously (*Gorwood et al., 2020*). The positive blue staining of SA-β-galactosidase has been used as a biomarker of cellular senescence. To detect SA-β-galactosidase activity, cells were incubated in an appropriate buffer solution at pH 6 containing bromo-4-chloro-3-indolyl-β-D-galactopyranoside (Euromedex, Souffelweyersheim, France), as described previously (*Gorwood et al., 2020*). The percentage of blue SA-β-galactosidase-positive cells was estimated by cell counting in at least three random-selected fields at a magnification of 20× . The acidotropic dye Lysotracker (Invitrogen Corporation, Carlsbad, CA) was used to evaluate the lysosomal mass. Cells cultured in 96-well plates (Corning, New York, NY) were incubated with Lysotracker for 2 hr at 37 °C. The fluorescence was quantified on a plate reader at 504 –570 nm (Tecan, Trappes, France), and normalized against 4′,6-diamidino-2-phenylindole dihydrochloride (DAPI) fluorescence at 345 –458 nm.

## Mitochondrial dysfunctions and oxidative stress

The cationic dye tetra-chloro-tetra-ethyl-benzimidazolyl-carbocyanine iodide (JC1) was used to evaluate the mitochondrial membrane potential, and the Mitotracker Red probe (both from Invitrogen Corporation) was used to measure the mitochondrial mass. The production of ROS was assessed by the oxidation of 5–6-chloromethyl-2,7-dichlorodihydrofluorescein diacetate (CM-H$_2$DCFDA) (Invitrogen Corporation). Cells cultured in 96-well plates were incubated with JC1, Mitotracker, or CM-H$_2$DCFDA for 2 hr at 37 °C. The fluorescence was quantified on a plate reader at 520 –595 nm for JC1 aggregates, 485 –535 nm for JC1 monomers, 575 –620 nm for Mitotracker, and 485 –520 nm for CM-H$_2$DCFDA. The results were normalized against DAPI fluorescence.

## Mitochondrial bioenergetics analysis

Cellular bioenergetic profiles were determined using the Seahorse XF Cell Mito Stress Test and the Bioscience XF24 Analyzer (Agilent), providing real-time measurement of the OCR. ASCs seeded or submitted to adipocyte differentiation in XF24 Microplates (2000 cells/well) were washed two times and incubated for 1 hr in FBS- and bicarbonate-free DMEM (pH 7.4) supplemented with 1 g/L (for ASCs) or 4.5 g/L (for adipocytes) glucose respectively, 1 % GlutaMax and 1 % pyruvate (Gibco). OCR was then evaluated using the Seahorse XF Cell Mito Stress Test (Agilent) at baseline and after sequential addition of 1 µmol/L oligomycin (inhibitor of ATP synthase), 0.63 µmol/L carbonyl cyanide FCCP

(uncoupling agent), and 1 µmol/L rotenone/antimycin A (R/AA, inhibitors of the respiratory chain complexes I and III, respectively).

## Protein extraction and western blotting

Proteins were extracted from cell monolayers as described previously (*Gorwood et al., 2019*; *Gorwood et al., 2020*). After SDS-PAGE, the proteins were transferred to nitrocellulose membranes. Specific proteins were detected using antibodies against p16INK4A, p21WAF1 (BD Bioscience, Franklin Lakes, NJ), prelamin A, PPARγ, CEBPα, SREBP1c, AMPK, phospho-AMPK (Cell Signaling Technology, Danvers, MA), and the protein loading control tubulin (Merck, Sigma-Aldrich). Immunoreactive complexes were detected using HRP-conjugated secondary antibodies (Cell Signaling Technology, Danvers, MA) and enhanced chemiluminescence (Thermo Fisher Scientific).

## Insulin signaling and glucose transport

The insulin sensitivity of adipocyte-derived ASCs at late passage (P11) was evaluated by the phosphorylation of Akt. On day 14, the adipocytes were serum-starved for 18 hr and stimulated or not for 7 min with 100 nmol/L insulin. Cell lysates were immunoblotted with antibodies against the activated forms (Ser473 phosphorylation) of Akt (Cell Signaling Technology). Protein expression was checked using antibodies against Akt (Cell Signaling Technology). Insulin-stimulated glucose uptake was measured using the Glucose Uptake-Glo assay kit (Promega, Fitchburg, WI), according to the manufacturer's instructions. Briefly, on day 14, adipocytes were serum-starved for 18 hr. Prior to the experiment, cells were incubated for 1 hr in glucose-free DMEM (Thermo Fisher Scientific). Next, 100 nmol/L insulin and 2-deoxyglucose mix were successively added for 60 and 10 min, respectively. In parallel with the insulin-stimulated and non-stimulated conditions, 50 µmol/L of the actin-disrupting agent cytochalasin B (Merck, Sigma-Aldrich) was added as a negative control and enabled determination of the net insulin-stimulated glucose uptake. The reaction was stopped with neutralization buffer and detection reagent mix was added and incubated for 1 hr at room temperature prior to measurement of luminescence on a plate reader.

## Statistical analysis

All experiments were performed in duplicate or triplicate on ASCs isolated from at least three or five different donors from each age group. All data were expressed as the mean ± SEM. The statistical significance of intergroup differences or changes over time was determined by applying a nonparametric test (the Mann–Whitney test), as appropriate.

## Acknowledgements

We thank the donors for their cooperation.

## Additional information

### Funding

| Funder | Grant reference number | Author |
|---|---|---|
| Fondation pour la Recherche Médicale | EQU201903007868 | Bruno Fève |
| Agence Nationale de la Recherche | RHU-ANR-15-RHUS-0003 | Jacqueline Capeau |

| Funder | Grant reference number | Author |
|---|---|---|
| Institut National de la Santé et de la Recherche Médicale | | Laura Le Pelletier<br>Matthieu Mantecon<br>Jennifer Gorwood<br>Martine Auclair<br>Roberta Foresti<br>Roberto Motterlini<br>Michael Atlan<br>Bruno Fève<br>Jacqueline Capeau<br>Claire Lagathu<br>Veronique Bereziat |
| Sorbonne University | | Laura Le Pelletier<br>Matthieu Mantecon<br>Jennifer Gorwood<br>Martine Auclair<br>Michael Atlan<br>Bruno Fève<br>Jacqueline Capeau<br>Claire Lagathu<br>Veronique Bereziat |
| Universite Paris Est Creteil | | Roberta Foresti<br>Roberto Motterlini |
| Universite de Paris | | Mireille Laforge |

The funders had no role in study design, data collection and interpretation, or the decision to submit the work for publication.

## Author contributions

Laura Le Pelletier, Conceptualization, Data curation, Formal analysis, Investigation, Methodology, Writing – original draft, Writing – review and editing; Matthieu Mantecon, Jennifer Gorwood, Conceptualization, Data curation, Formal analysis, Investigation, Methodology; Martine Auclair, Data curation, Formal analysis, Investigation, Methodology; Roberta Foresti, Formal analysis, Methodology, Validation; Roberto Motterlini, Methodology, Validation; Mireille Laforge, Methodology; Michael Atlan, Resources; Bruno Fève, Jacqueline Capeau, Funding acquisition, Writing – review and editing; Claire Lagathu, Conceptualization, Data curation, Funding acquisition, Investigation, Methodology, Project administration, Supervision, Writing – original draft, Writing – review and editing; Veronique Bereziat, Conceptualization, Formal analysis, Funding acquisition, Investigation, Methodology, Resources, Supervision, Validation, Writing – original draft, Writing – review and editing

## Author ORCIDs

Veronique Bereziat http://orcid.org/0000-0002-9795-549X

## Decision letter and Author response
Decision letter https://doi.org/10.7554/eLife.62635.sa1
Author response https://doi.org/10.7554/eLife.62635.sa2

# Additional files

## Supplementary files
• Transparent reporting form

## Data availability
All data generated or analysed during this study are included in the manuscript and supporting files.

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
