## [Decision Letter]

**Acceptance summary:**

The revised manuscript has been greatly improved. The authors provided more consolidated evidence linking fundamental aging processes, oxidative stress, cellular senescence and mitochondrial dysfunction to a progressive functional decline in adipose tissue. They also performed more mechanistic studies to explain the beneficial effects of metformin, such as stress resistance in human ASC. I believe that the authors provide sufficient data to support the main hypothesis in the current manuscript. I'm satisfied with all responses from the authors.

**Decision letter after peer review:**

Thank you for submitting your article "Metformin alleviates aging-associated cellular senescence of human adipose stem cells and derived adipocytes" for consideration by *eLife*. Your article has been reviewed by 2 peer reviewers, and the evaluation has been overseen by a Reviewing Editor and Jessica Tyler as the Senior Editor. The following individual involved in review of your submission has agreed to reveal their identity: Yi Zhu (Reviewer #1).

The reviewers have discussed the reviews with one another and the Reviewing Editor has drafted this decision to help you prepare a revised submission.

We feel that the major conclusions are right but the manuscript and story is not quite clear enough at present and there is a lack of deeper cellular and molecular mechanistic understanding of these phenomena to distinguish this work from the previous published studies. That ASC senescence impairs adipogenic differentiation capacity, has been previously reported in e*Life* in 2015 (doi: 10.7554/*eLife*.12997). For example, you concluded that adipose derived progenitor cells from older adults have higher potential to become senescence, which impaired adipogenesis. The percentage of senescent cells in adipose tissues is low, but the mechanism of how they could significantly affect adipose tissue functions is unknown. Is this through paracrine effects? or cross talk with other immune cells? etc..

Essential revisions:

1. Although the authors found that ASC senescence is associated with mitochondrial dysfunction and oxidative stress, it the nature of the links between these cellular events is unclear. It is well-known that mitochondrial dysfunction can directly lead to senescence. If the authors meant to prove that ASC senescence causes early adipocyte mitochondrial dysfunction, more evidence is required.

2. It has already been reported that ASC senescence impairs adipogenic differentiation capacity, in e*Life* in 2015 (doi: 10.7554/*eLife*.12997). Furthermore, although the authors found that metformin prevents the onset of senescence and associated dysfunctions in ASCs, it has been shown in many publications that metformin is a senomorphic drug that can reduce the senescence-associated secretory phenotype. So it is not surprising that metformin can block the effects of senescent ASCs. Also regarding the increased adipogenesis by metformin, it has been reported that metformin can directly regulate adipogenic transcription factors, such as peroxisome proliferator-activated receptor (PPARγ), CCAAT/enhancer binding protein α (C/EBPα). As such, sufficient novelty is lacking at this point, and would require demonstration of causal links among these cellular events.

3. Several conclusions need to be smoothed out and discussed in more detail. Methods must be described with more details, especially with regard to fat depot digestion (type of collagenase, concentration of collagenase, amount of tissue used for the digestion, are cell yields similar between young and old adipose tissue? Number of plated ASC?). The authors must consider that the term ASC is nowadays related to Adipose stromal cells and not Stem cells. As described in the introduction and method sections, ASC are stromal cells that adhere to plastic including fibroblast, smooth muscle cells, pericytes, endothelial cells, resident macrophages, preadipocytes and progenitors. This must be discussed since distribution and repartition of stromal cells are modulated with aging. The term "adipocyte" must be changed to "differentiated ASC" because adipocytes are characterized by unique lipid droplet (not the case here). The title must be modified. Senescence is related to ASC and not to adipocytes.

4. Figure 1: It is unclear why the authors conclude about they are recapitulating in vivo aging. If so, one might expect that senescent "young ASC" phenotype may recapitulate the one of "old ASC" with a time lag, what is not the case for all the studied parameters. For example, the % of βGal cells is equivalent between P7 old cells and P11 young cells what is also true for P16, P21 and prelamin A but not for reactive oxygen species or mitochondrial potential. Authors must discuss this point.

5. Figure 2: Was Cell number at confluency controlled and similar between "young" and "old" ASC? Since post-confluent mitosis are necessary for adipogenesis, one might speculate that the decreased adipogenesis might be related to less cell number and proliferation.

6. Figure 3 and 4: Cells were treated from P3 with metformin. Do the authors consider potential "resistance" effect? When taking into account the large number of individuals treated with metformin, is there any evidence of an impact of metformin treatment on age-related loss in subcutaneous adipose tissue? Finally, inhibition of senescence may lead to cancer development. Authors must discuss this point.

---

## [Author Response]

We feel that the major conclusions are right but the manuscript and story is not quite clear enough at present and there is a lack of deeper cellular and molecular mechanistic understanding of these phenomena to distinguish this work from the previous published studies. That ASC senescence impairs adipogenic differentiation capacity, has been previously reported in eLife in 2015 (doi: 10.7554/eLife.12997). For example, you concluded that adipose derived progenitor cells from older adults have higher potential to become senescence, which impaired adipogenesis. The percentage of senescent cells in adipose tissues is low, but the mechanism of how they could significantly affect adipose tissue functions is unknown. Is this through paracrine effects? or cross talk with other immune cells? etc..

We would like to thank the reviewers, and the reviewing Editor for their constructive comments that helped us to improve the quality of our manuscript. We now clearly outline that ASCs senescence could be either related to the in vitro aging (due to increased cell passages) or to the in vivo aging (stress-induced). These situations differently affect adipocyte differentiation and function. In our model we could not evaluate the role of the cross-talk with immune cells. We indicate that point as a limitation at the end of the discussion. We confirm the potential role of activin secreted by senescent cells and acting through paracrine mechanisms as previously shown [1]. We provide below a point-by-point reply to the reviewers’ remarks and criticisms.

Essential revisions:1. Although the authors found that ASC senescence is associated with mitochondrial dysfunction and oxidative stress, it the nature of the links between these cellular events is unclear. It is well-known that mitochondrial dysfunction can directly lead to senescence.

In this study we used ASCs isolated from subcutaneous adipose tissue of young and aged donors (named young-donor ASCs and aged-donor ASCs) both of which underwent in vitro aging over the culture passages. Therefore, we can evaluate 3 age-related processes: in vitro aging alone (young-donor ASCs with increasing passage number), in vivo physiological aging alone (aged-donor ASCs at early passage, P3) and in vitro and in vivo aging together (aged-donor ASCs at increasing cell passage numbers). Thus, we were able to analyze the defects presented by ASCs in each situation and the resulting adipose dysfunction presented by adipocyte-differentiated ASCs.

Regarding young-donor ASCs, we observed that, at P3, the level of senescence was minimal but increased cell passage number in vitro led to ASCs senescence without oxidative stress. In this situation, adipocyte differentiated-ASCs showed decreased adipogenesis starting from P7 but without mitochondrial dysfunction, despite mild oxidative stress.

Conversely, physiological aging (aged-donor ASCs at P3) was characterized by oxidative stress in proliferating ASCs but no mitochondrial dysfunction as evaluated by JC1. To go further, we performed additional Seahorse experiments. Young- and aged-donor ASCs presented a similar respiration profile, indicating the absence of mitochondrial dysfunction. These results suggest that increased ROS production observed in aged-donor ASCs at P3 did not initially result from mitochondrial dysfunction (new Figure 2). Interestingly, adipocytes differentiated from these aged ASCs, at this early passage, maintained their adipogenic potential but presented early oxidative stress. In addition, they presented mitochondrial dysfunction as outlined by seahorse experiments (new Figure 4). Altogether, these results are in favor of different senescence mechanisms depending on chronological (in vitro passages) or physiological (age of the donor) aging. in vitro passageinduced senescence impedes adipocyte differentiation while stress-induced senescence (as observed in physiological aging in aged-donor ASCs at P3) results in oxidative stress, mitochondrial dysfunction but not impaired differentiation in derived adipocytes. When aged-donor ASCs age in vitro, the two processes are present resulting in severe dysfunction in adipocyte differentiated-ASCs.

We have now changed the manuscript accordingly in the results and Discussion sections and added the seahorse experiment (Figure 2 and 4 in the revised manuscript). For these experiments, we benefited from the help and expertise of our collaborators Roberta Foresti, Roberto Motterlini, and Mireille Laforge. Therefore, they were included in the authors of the article.

If the authors meant to prove that ASC senescence causes early adipocyte mitochondrial dysfunction, more evidence is required.

Our data reveal that in vitro aging of young-donor ASCs, that resulted in ASC senescence, was not associated with early mitochondrial dysfunction in derived adipocytes (at P7, Figure 3F and G), thus disentangling ASC senescence and adipocyte mitochondrial dysfunction. This sentence was added in the Results section.

To investigate the impact of physiological aging of ASCs, we analyzed the respiration profile at P3 in adipocyte differentiated-ASCs from young and aged donors. The analysis of basal OCR did not reveal any significant difference. However, the adipocyte differentiated from aged-donors ASCs displayed a major decrease in their maximal respiratory capacity as compared to those from young-donor ASCs, as indicated by a significantly lower increase in OCR after the addition of the uncoupling agent FCCP (new Figure. 4). Therefore, increased oxidative stress associated with mild senescence in aged-donor ASCs resulted in mitochondrial dysfunction in ASC-derived adipocytes, suggesting that adipocyte mitochondrial dysfunction could be linked to increased oxidative stress in ASCs. This sentence was added in the Result section and discussed in the Discussion section.

To further delineate the link between oxidative stress and senescence of ASCs and of derived adipocytes, we induced senescence of young-donor ASCs at P3 by treating them with an HIV protease inhibitor (PI) and analyzed the impact of this stress-induced senescence on adipocyte differentiated-ASCs. PIs are known to induce senescence by accumulating farnesylated prelamin A, responsible for premature aging syndromes [2-5]. As shown in new Figure 5, a treatment for 30 days of ASCs with lopinavir associated with a low dose of ritonavir (LPV/r) led to senescence but also to a high level of oxidative stress and mitochondrial dysfunctions (destabilization of the mitochondrial membrane potential, as shown by the JC1 assay despite increased mitochondrial mass). Adipocytes derived from these LPV/r-treated ASCs, which presented severe oxidative stress and marked senescence, exhibited increased oxidative stress and mitochondrial dysfunctions together with altered adipogenesis. This confirmed that induction of oxidative stress and senescence in young ASCs resulted in severe adipose dysfunction in derived adipocytes. Interestingly, another π with less adverse effects, darunavir associated with a low dose of ritonavir (DRV/r), did not increase senescence in ASCs but induced mild oxidative stress and mitochondrial dysfunction. Accordingly, treatment of ASCs with DRV/r did not impede adipocyte differentiation, further indicating that mild oxidative stress in the absence of senescence in ASCs did not impair adipogenesis.

Overall, our results are in favor of the hypothesis that mild oxidative stress and mitochondrial dysfunction do not preclude differentiation of ASC-derived adipocyte. Therefore, we were able to disentangle the effect of senescence alone versus oxidative stress on ASC-derived adipocyte differentiation, oxidative stress and mitochondrial dysfunction. We added these data in the revised manuscript with new figures 2, 4, and 5.

2. It has already been reported that ASC senescence impairs adipogenic differentiation capacity, in eLife in 2015 (doi: 10.7554/eLife.12997).

We agree with the reviewer that it has already been reported that ASC senescence impairs adipogenic differentiation capacity. Our data are in agreement with this point. Moreover, we further show that in vitro passage-related senescence impairs adipogenesis even in the absence of increased oxidative stress. Thus, in our study we show, for the first time, that during physiological aging (aged-donor ASCs at early passage), adipocyte differentiated-ASCs present early oxidative stress and mitochondrial dysfunctions but not reduced adipogenic potential. Thus, senescence, but not oxidative stress alone, impairs adipogenic differentiation. However, when oxidative stress is high, this results in stress-induced senescence and in turn in reduced adipogenic differentiation. Our results allow to better decipher the respective role of oxidative stress and senescence regarding adipogenesis.

Furthermore, although the authors found that metformin prevents the onset of senescence and associated dysfunctions in ASCs, it has been shown in many publications that metformin is a senomorphic drug that can reduce the senescence-associated secretory phenotype. So it is not surprising that metformin can block the effects of senescent ASCs.

Regarding the effect of metformin, we agree with the reviewer that it can reduce the SASP. Accordingly, we observed a decrease in activin A secretion in response to metformin in both young-donor and aged-donor ASCs regardless the passage (Author response image 1). However, in our model metformin did not restore the adipogenic capacity of young-donor ASCs at P11 that present senescence but not oxidative stress, suggesting that (i) metformin could act by another mechanism and (ii) metformin was not able to reverse the ASCs’ senescence features when oxidative stress is low (iii) metformin was able to reverse stress-induced senescence by acting on oxidative stress.

**Author response image 1. sa2fig1:** Activin A secretion increases with ASCs senescence. The results correspond to the mean ± SEM. *P<0.05, **P<0.01, ***P<0.001 for aged-donor vs. young-donor ASCs, §P<0.05, §§P<0.01 metformin-treated vs. non-treated ASCs. All experiments were performed in duplicate in ASCs isolated from 3 different donors in each group. ^#^P<0.05, ^##^P<0.01, ^###^P<0.001 vs. young- or aged-donor ASCs at P3.

As mentioned in the Discussion section, previous studies have demonstrated that metformin significantly improved ASC proliferation and function, which were correlated with a higher mitochondrial membrane potential [6]. Accordingly, we found that metformin was a potent antioxidant and prevented the onset of mitochondrial dysfunction in aged-donor ASCs. In young-donor ASCs at P11, which display senescence features and low oxidative stress, metformin had no impact on oxidative stress and did not improve adipogenesis.

This effect is believed to be related to metformin’s ability to inhibit complex 1 in the mitochondrial electron transport chain, leading to activation of AMPK. In agreement, we observed that the beneficial effects of metformin were lost in the presence of compound C (a potent AMPK inhibitor) (see figure 8 in the revised manuscript) – highlighting the central role of the AMPK-linked signaling network in the ASCs aging process [7,8]. To confirm this hypothesis, we directly activated AMPK using AICAR (5-aminoimidazole-4-carboxyamide ribonucleoside), an agonist of the AMPK pathway [9]. The direct activation of AMPK by AICAR led to an impact on ASCs similar to that observed in the presence of metformin, characterized by decreased oxidative stress and senescence of aged-donor ASCs at P11 (new Figure 8—figure supplement 1). AICAR treatment also led to decreased oxidative stress and mitochondrial dysfunctions in derived adipocytes together with improved adipogenesis (new Figure 8—figure supplement 2).

Also regarding the increased adipogenesis by metformin, it has been reported that metformin can directly regulate adipogenic transcription factors, such as peroxisome proliferator-activated receptor (PPARγ), CCAAT/enhancer binding protein α (C/EBPα). As such, sufficient novelty is lacking at this point, and would require demonstration of causal links among these cellular events.

As mentioned by the reviewer, metformin can directly regulate adipogenic transcription factors. Metformin has been shown to inhibit [10-12] or increase adipogenesis [13], depending on the concentration used [14], when cells are exposed throughout the differentiation process. In our study, we used a different model in which ASCs were treated with metformin only before the induction of differentiation. By removing metformin during ASCs differentiation, we were able to rule out a direct effect on adipogenesis (new Figure 7—figure supplement 1). We have added a figure illustrating the protocol used in our study to clarify this point.

3. Several conclusions need to be smoothed out and discussed in more detail.

We believe that all the constructive comments made by the reviewers and editor improved the quality of our manuscript. In addition to all the changes made throughout the paper, we toned down some of our conclusions. Moreover, our results allowed us to discuss the fact that ASCs senescence could be either related to in vitro aging (due to increased cell passages) or to in vivo aging (stress-induced).

Methods must be described with more details, especially with regard to fat depot digestion (type of collagenase, concentration of collagenase, amount of tissue used for the digestion, are cell yields similar between young and old adipose tissue? Number of plated ASC?).

We agree with the reviewer and we added the missing information in the Material and Methods section. The amount of tissue used depended on the volume of the sample withdrawn during surgery. However, the collagenase digestion protocol was always the same (we used a ratio of 2/3 of 0,2% collagenase combined with 1/3 of adipose tissue). Similarly, the cells were counted and plated at the same density (10,000 cells/cm^2^ in T75 flasks). On average, we isolated 65,000 cells per mL of SCAT liposuction sample, with similar yield in SCAT obtained from aged or young donors. These points are now detailed in the Materials and methods section.

The authors must consider that the term ASC is nowadays related to Adipose stromal cells and not Stem cells. As described in the introduction and method sections, ASC are stromal cells that adhere to plastic including fibroblast, smooth muscle cells, pericytes, endothelial cells, resident macrophages, preadipocytes and progenitors. This must be discussed since distribution and repartition of stromal cells are modulated with aging.

As requested we have replaced the term “stem cell” by “stromal cell “in the manuscript.

The term "adipocyte" must be changed to "differentiated ASC" because adipocytes are characterized by unique lipid droplet (not the case here).

In our study, we use an in vitro model that leads, as mentioned by the reviewer, to adipocyte with multilocular lipid droplet. This is always the case in vitro. We changed the term “adipocyte to “adipocyte differentiated from ASCs” or ‘ACS-derived adipocytes” throughout the manuscript.

The title must be modified. Senescence is related to ASC and not to adipocytes.

We have changed the title for “Metformin alleviates stress-induced cellular senescence of aging human adipose stromal cells and the ensuing adipocyte dysfunction”

4. Figure 1: It is unclear why the authors conclude about they are recapitulating in vivo aging. If so, one might expect that senescent "young ASC" phenotype may recapitulate the one of "old ASC" with a time lag, what is not the case for all the studied parameters. For example, the % of βGal cells is equivalent between P7 old cells and P11 young cells what is also true for P16, P21 and prelamin A but not for reactive oxygen species or mitochondrial potential. Authors must discuss this point.

We agree with the reviewer and we have modified the text and distinguished different types of aging as indicated above. In our study in vitro increased cell passage numbers led to young-donor ASCs senescence without oxidative stress. Adipocyte differentiated from young ASCs showed decreased adipogenesis starting from P7 but without mitochondrial dysfunction. Conversely, physiological aging (aged-donors versus young-donors) led to premature senescence and increased oxidative stress as soon as P3 as observed in proliferating ASCs obtained from aged-donors. Interestingly, the adipocyte differentiated-aged ASCs maintained their adipogenic potential but presented early mitochondrial dysfunction and oxidative stress at P3.

In addition, we found that metformin was a potent antioxidant and prevented the onset of mitochondrial dysfunction only in aged-donor ASCs. In young-donor ASCs at P11, metformin has no impact on oxidative stress and doesn’t improve adipogenesis. Altogether, these results are in favor of different senescence mechanisms depending on chronological (in vitro passages) or physiological (age of the donor) aging. We have now extensively changed the manuscript in both the Results and Discussion sections.

5. Figure 2: Was Cell number at confluency controlled and similar between "young" and "old" ASC? Since post-confluent mitosis are necessary for adipogenesis, one might speculate that the decreased adipogenesis might be related to less cell number and proliferation.

As requested by the reviewer, we determined post-confluent cell number just before the induction of the differentiation process and observed no difference between young-donor and aged-donor ASCs. Moreover, ASCs were seeded in 6-well plates at a high density, thus preventing any delay between young- and aged-donor derived ASCs.

**Author response image 2. sa2fig2:** Post confluent cell number in young-donor and aged-donor ASCs.

6. Figure 3 and 4: Cells were treated from P3 with metformin. Do the authors consider potential "resistance" effect?

We used this protocol to determine if metformin could prevent the onset of senescence. However, we performed an additional experiment in which AMPK was directly activated by its agonist AICAR. In these experiments, young-donor and aged-donor ASCs were cultured until P11 and then treated or not with AICAR and metformin for 7 days. Our results clearly show that AMPK activation reversed ASCs senescence and led to functional adipocytes. Since the effect of AICAR was observed after 7 days of treatment and was similar to the effect of metformin given for a long period, we think that our results cannot be explained by a resistance to metformin after a long duration of treatment.

When taking into account the large number of individuals treated with metformin, is there any evidence of an impact of metformin treatment on age-related loss in subcutaneous adipose tissue?

Several clinical studies strongly suggest that metformin could improve adipose tissue function. A treatment with metformin leads, in most cases, to reduced weight, i.e. global fat loss, and improved insulin sensitivity, and these effects are largely related to actions outside adipose tissue. A few studies evaluated the effect of metformin on the different fat depots. In the large study “Diabetes Prevention Program Research group” with more than 3,000 glucose intolerant subjects, mean age 54 years, treated with placebo or metformin or lifestyle intervention, both SCAT and VAT were reduced in women after 1 year of treatment with metformin [15]. As well, adiponectin level was mildly improved while leptin was decreased in favor of improved adipose tissue function [16]. Improved adipose tissue function is also observed in SCAT explants issued from obese subjects treated ex vivo with metformin. The treatment resulted in increased adiponectin expression. As well, a 3-month treatment of obese subjects with metformin resulted in increased adiponectin expression in SCAT. Interestingly this effect was observed in SCAT biopsies but not when adipocytes differentiated from patients’ adipose stromal cells were treated in vitro with metformin suggesting that metformin affected cells from the stroma vascular fraction, that contains adipose stromal cells, and not directly differentiated adipocytes [17]. Finally, a study evaluated, in a cross-over study, the effect of metformin versus placebo on SCAT parameters from aged subjects (mean age 71 years). As expected, treatment with metformin resulted in weight loss and overall improved insulin sensitivity. Interestingly, in adipose tissue, metformin improved expression of genes involved in PPAR signaling and in SREBP signaling and triglyceride synthesis, arguing for its ability to improve adipose function during aging [18].

Part of this paragraph is now added in the Discussion section.

Finally, inhibition of senescence may lead to cancer development. Authors must discuss this point.

Our data show that metformin did not improve directly senescence of adipose stromal cells but decreased oxidative stress and therefore decreased stress-induced senescence in these cells. Reactive oxygen species are considered as favoring DNA damage and carcinogenesis. To reduce oxidative stress is generally considered as favorable regarding this process.

Regarding the ability of metformin to modulate cancer development, a high number of studies have shown that metformin exerts antitumor effects. At first, the ability of metformin to reduce cancer risk was reported in patients with type-2 diabetes receiving metformin as an antidiabetic agent. Then, this was also shown in retrospective and meta-analysis studies regarding some cancers, which showed that metformin improved recurrence-free survival and overall survival, mainly in early-stage colorectal and prostate cancers [19]. Different mechanisms have been described among which the activation of AMPK and the reduction of oxidative stress but also a number of mechanisms independent of AMPK activation [20]. Even if studies are required to further decipher these mechanisms, no alert on a pro-oncogenic effect of metformin has been raised.

References:

1. Xu M. et al., *eLife* 2015. DOI: 10.7554/*eLife*.12997

2. Afonso P. et al., Antivir Ther 2017. DOI: 10.3851/IMP3160

3. Afonso P. et al., Atherosclerosis 2016. DOI: 10.1016/j.atherosclerosis.2015.12.012

4. Hernandez-Vallejo SJ. et al., Aging Cell 2013. DOI: 10.1111/acel.12119

5. Lefevre C. et al., Arterioscler Thromb Vasc Biol. 2010. DOI:10.1161/ATVBAHA

6. Smieszek A. et al., Cells 2019. DOI:10.3390/cells8020080

7. Salminen A. et al., Ageing Res Rev 2012. DOI: 10.1016/j.arr.2011.12.005

8. Burkewitz K. et al., Cell Metab 2014. DOI: 10.1016/j.cmet.2014.03.002

9. Merrill GF. et al., Am J Physiol 1997. DOI: 10.1152/ajpendo.1997.273.6.E1107

10. Chen SC. et al., Mol Cell Endocrinol 2017. DOI: 10.1016/j.mce.2016.11.011

11. Gu Q. et al., Exp Med 2017. DOI: 10.1620/tjem.241.13.

12. Marycz K. et al., Oxid Med Cell Longev 2016. DOI: 10.1155/2016/9785890

13. Jaganjac M. et al., Redox Biol 2017. DOI: 10.1016/j.redox.2017.03.012

14. Chen D. et al., Int J Mol Sci 2018. DOI: 10.3390/ijms19061547

15. Fujimoto W. et al., Diabetes 2007. DOI: 10.2337/db07-0009

16. Goldberg R. et al., Diabetologia 2019. DOI: 10.1007/s00125-018-4748-2

17. Zulian A. et al., Obes Facts 2011. DOI: 10.1159/000324582

18. Kulkarni AS. et al., Aging Cell 2018. DOI: 10.1111/acel.12723

19. Coyle C. et al., Ann Oncol 2016. DOI: 10.1093/annonc/mdw410

20. Mbara KC. et al., Eur J Pharmacol 2021.DOI: 10.1016/j.ejphar.2021.173934

21. Ragnauth CD. et al., Circulation, 2010. DOI: 10.1161/CIRCULATIONAHA.109.902056

22. Bonello-Palot N. et al., Atherosclerosis 2014. DOI: 10.1016/j.atherosclerosis.2014.08.036

23. Bidault G. et al., Cells 2020. DOI: 10.3390/cells9051201